# A general decoding strategy explains the relationship between behavior and correlated variability

Amy M Ni[1,2]*, Chengcheng Huang[1,2,3], Brent Doiron[2,3], Marlene R Cohen[1,2]

[1]Department of Neuroscience,University of Pittsburgh, Pittsburgh, United States; [2]Center for the Neural Basis of Cognition, Pittsburgh, United States; [3]Department of Mathematics, University of Pittsburgh, Pittsburgh, United States

**Abstract** Improvements in perception are frequently accompanied by decreases in correlated variability in sensory cortex. This relationship is puzzling because overall changes in correlated variability should minimally affect optimal information coding. We hypothesize that this relationship arises because instead of using optimal strategies for decoding the specific stimuli at hand, observers prioritize *generality*: a single set of neuronal weights to decode any stimuli. We tested this using a combination of multineuron recordings in the visual cortex of behaving rhesus monkeys and a cortical circuit model. We found that general decoders optimized for broad rather than narrow sets of visual stimuli better matched the animals' decoding strategy, and that their performance was more related to the magnitude of correlated variability. In conclusion, the inverse relationship between perceptual performance and correlated variability can be explained by observers using a general decoding strategy, capable of decoding neuronal responses to the variety of stimuli encountered in natural vision.

## Editor's evaluation

Empirical findings have established that experimental manipulations which increase perceptual accuracy also generally reduce the amount of shared variability between neurons in the visual cortex. To explain this observation, this study combines neurophysiology data and a network model of visual cortex and tests the hypothesis that perception relies on a "general" decoding strategy. The results suggest that the brain seeks to decode arbitrary changes in stimuli that appear in the environment.

*For correspondence: amn75@pitt.edu

Competing interest: The authors declare that no competing interests exist.

## Introduction

Many studies have demonstrated that increases in perceptual performance correspond to decreases in a very simple measure of shared variability in a population of sensory neurons: the mean correlation between the responses of a pair of neurons to repeated presentations of the same stimulus (termed spike count or noise correlations, or $r_{SC}$; *Cohen and Kohn, 2011*; *Nirenberg and Latham, 2003*; *Cohen and Maunsell, 2009*; *Cohen and Maunsell, 2011*; *Gregoriou et al., 2014*; *Gu et al., 2011*; *Herrero et al., 2013*; *Luo and Maunsell, 2015*; *Mayo and Maunsell, 2016*; *Mitchell et al., 2009*; *Nandy et al., 2017*; *Ni et al., 2018*; *Ruff and Cohen, 2014a*; *Ruff and Cohen, 2014b*; *Ruff and Cohen, 2016*; *Ruff and Cohen, 2019*; *Yan et al., 2014*; *Zénon and Krauzlis, 2012*). We recently found that the axis in neuronal population space that explains the most mean correlated variability explains virtually all of the choice-predictive signals in visual area V4 (*Ni et al., 2018*).

These observations comprise a paradox because changes in this simple measure should have a minimal effect on information coding. Recent theoretical work shows that neuronal population

decoders that extract the maximum amount of sensory information for the specific task at hand can easily ignore mean correlated noise (*Kafashan et al., 2021*; *Kanitscheider et al., 2015b*; *Moreno-Bote et al., 2014*; *Pitkow et al., 2015*; *Rumyantsev et al., 2020*; for review, see *Kohn et al., 2016*). Decoders for the specific task at hand can ignore mean correlated variability because it does not corrupt the dimensions of neuronal population space that are most informative about the stimulus (*Moreno-Bote et al., 2014*).

We propose a hypothesis that reconciles the numerous experimental observations of an inverse relationship between performance and mean correlated variability and these seemingly contradictory theoretical predictions. The theoretical predictions are predicated on the assumption that observers (and their neuronal population decoding mechanisms) use a decoding strategy that maximizes the amount of information extracted for the specific task at hand. We hypothesize that instead, observers use a *general* decoding strategy: one set of neuronal population decoding weights to extract sensory information about any visual stimuli.

Stimuli in the visual environment vary in many task-irrelevant as well as -relevant features. Our idea is that the dimensions of the neural code that are optimal for the general decoding of any stimuli might be very similar to the axes that account for the mean correlated variability, because mean correlated variability is well known to depend on all of the stimulus features for which the population of neurons is tuned (*Cohen and Kohn, 2011*). If observers used this kind of general decoding strategy, their perceptual performance might be inextricably linked to mean correlated variability (*Ruff et al., 2018*). This decoding mechanism would explain the much-observed relationship between behavior and correlated variability.

Here, we report the results of an initial test of this overarching hypothesis, based on a single stimulus dimension. We used a simple, well-studied behavioral task to test whether a more-general decoder (optimized for a broader range of stimulus values along a single dimension) better explained the relationship between behavior and mean correlated variability than a more-specific decoder (optimized for a narrower range of stimulus values along a single dimension). Specifically, we used a well-studied orientation change detection task (*Cohen and Maunsell, 2009*) to test whether a general decoder for the full range of stimulus orientations better explained the relationship between behavior and mean correlated variability than a specific decoder for the orientation change presented in the behavioral trial at hand.

This test based on a single stimulus dimension is an important initial test of the general decoder hypothesis because many of the studies that found that performance increased when mean correlated variability decreased used a change detection task (*Cohen and Maunsell, 2009*; *Cohen and Maunsell, 2011*; *Herrero et al., 2013*; *Luo and Maunsell, 2015*; *Mayo and Maunsell, 2016*; *Nandy et al., 2017*; *Ni et al., 2018*; *Ruff and Cohen, 2016*; *Ruff and Cohen, 2019*; *Yan et al., 2014*; *Zénon and Krauzlis, 2012*). This task has been studied frequently because it is a simple laboratory version of a real-life scenario: the observer must report that a stimulus changed, regardless of the magnitude of the change.

For this test, we used a combination of experiments and theory. We used multineuron recordings in two rhesus monkeys to directly compare the effects of modulating visual attention on the mean correlated variability of the neuronal population to the effects of attention on the monkeys' neuronal decoding strategy. We then used a cortical circuit model to compare the effects of attention on the monkeys' decoding strategy to the effects of attention on an ideal general decoder for all orientations.

Our combined electrophysiological and theoretical results support our general decoder hypothesis. They demonstrate that using a single set of neuronal weights to decode sensory neuron population responses to any stimulus change can explain the frequently observed relationship between performance and mean correlated variability.

## Results
### A behavioral framework for studying the general decoder hypothesis

We designed a behavioral task for two rhesus monkeys that allowed us to test the hypothesis that the relationship between perceptual performance and correlated variability is better explained by a more-general decoding strategy. This test required two main components. First, we used an orientation change detection task with multiple potential orientation changes (*Figure 1A*; different aspects of

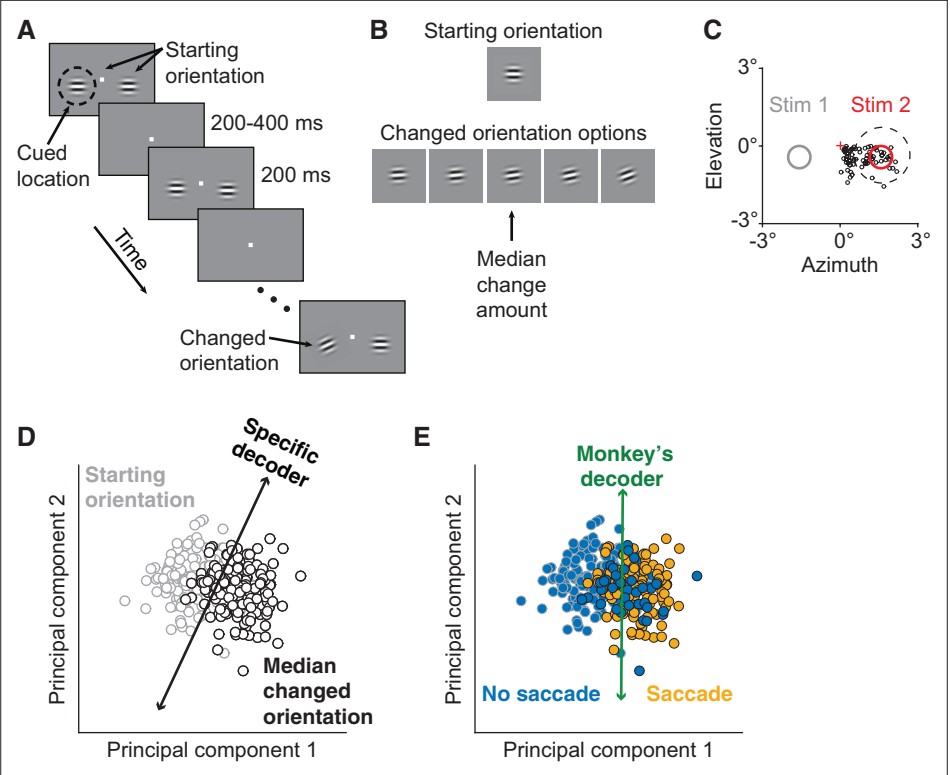

**Figure 1.** Electrophysiological data collection and decoders. (**A**) Orientation change detection task with cued attention. After the monkey fixated the central spot, two Gabor stimuli synchronously flashed on (200 ms) and off (randomized 200–400 ms period) at the starting orientation until, at a random time, the orientation of one stimulus changed. To manipulate attention, the monkey was cued in blocks of 125 trials as to which of the two stimuli would change in 80% of the trials in the block, with the change occurring at the uncued location in the other 20%. (**B**) A cued changed orientation was randomly assigned per trial from five potential orientations. An uncued changed orientation was randomly either the median (20 trials) or largest change amount (5 trials). To compare cued to uncued changes, median orientation change trials were analyzed. (**C**) The activity of a neuronal population in V4 was recorded simultaneously. Plotted for Monkey 1: the location of Stimulus 2 (red circle) relative to fixation (red cross) overlapped the receptive field (RF) centers of the recorded units (black circles). A representative RF size is illustrated (dashed circle). Only orientation changes at the RF location were analyzed. Stimulus 1 was located in the opposite hemifield (gray circle). (**D**) Schematic of the specific decoder, a linear classifier with leave-one-out cross-validation, which was trained to best differentiate the V4 neuronal population responses to the median changed orientation from the V4 responses to the starting orientation presented immediately before it (first and second principal components [PC] shown for illustrative purposes). (**E**) Schematic of the monkey's decoder, which was based on the same neuronal population responses as in (**D**) but was trained to best differentiate the V4 responses when the monkey made a saccade (indicating it detected an orientation change) from the V4 responses when the monkey did not choose to make a saccade.

these data were presented previously, *Ni et al., 2018*). This allowed us to analyze decoders optimized for narrower versus broader ranges of stimulus orientations. Two Gabor stimuli of the same orientation flashed on and off until, at a random time, the orientation of one of the stimuli changed. The changed orientation was randomly selected from five options (*Figure 1B*). The monkey could not predict which orientation change was to be detected on any given trial and was rewarded for responding to any orientation change.

Second, we made a manipulation designed to create a larger dynamic range of perceptual performance. We manipulated visual attention within the task (*Figure 1A*) using a classic Posner cueing paradigm (*Posner, 1980*). Cued trials were collected for all five change amounts and uncued trials were collected mainly for the median change amount (*Figure 1B*). Our attention analyses focused on this median change amount, for which we had both cued and uncued trials.

For each monkey, we used a chronically implanted microelectrode array to record from a population of V4 neurons while the monkey performed the behavioral task (*Figure 1C*). These electrophysiological recordings allowed us to measure the effects of attention on the mean correlated variability of the V4 population. They also allowed us to measure the effects of attention on the performance of two linear decoders of the V4 population activity: a specific decoder (*Figure 1D*) and the monkey's decoder (*Figure 1E*).

We tasked both decoders with differentiating the V4 neuronal population responses to the median changed orientation from the V4 responses to the starting orientation presented immediately before it. We first estimated the neuronal decoding weights that best performed this specific task (*Figure 1D*). Theoretical studies have found that mean correlated variability should not affect the performance of such an optimal linear decoder (*Kafashan et al., 2021*; *Kanitscheider et al., 2015b*; *Moreno-Bote et al., 2014*; *Pitkow et al., 2015*; *Rumyantsev et al., 2020*; reviewed by *Kohn et al., 2016*).

To compare the monkey's strategy to the performance of the specific decoder, we estimated the neuronal decoding weights that best predicted the monkey's pattern of choices on this same task (*Figure 1E*). This allowed us to directly compare the performance of the monkey's decoder to that of the specific decoder on the same task. Using leave-one-out cross-validation, we calculated the ability of each decoder to correctly identify whether each left-out orientation was the median changed orientation or the starting orientation.

## A mechanistic circuit model to test the general decoder hypothesis

Here, we describe a circuit model that we designed to allow us to compare the specific and monkey's decoders from our electrophysiological dataset to modeled ideal specific and general decoders. The primary benefit of our model is that it can take actual images as inputs and produce neuronal tuning and covariance that are compatible with each other because of constraints from the simulated network that processed the inputs (*Huang et al., 2019*). Parametric models in which tuning and covariance can be manipulated independently would not provide such constraints. In our model, the mean correlated variability of the population activity is restricted to very few dimensions, matching experimentally recorded data from visual cortex demonstrating that mean correlated variability occupies a low-dimensional subset of the full neuronal population space (*Ecker et al., 2014*; *Goris et al., 2014*; *Huang et al., 2019*; *Kanashiro et al., 2017*; *Lin et al., 2015*; *Rabinowitz et al., 2015*; *Semedo et al., 2019*; *Williamson et al., 2016*).

For our electrophysiological dataset, the behavioral task was designed to allow us to compare the specific and monkey's decoders for an attention task with a range of orientation change amounts. To collect the necessary number of repetitions of behavioral trials per stimulus condition (with the limited total number of behavioral trials collected per day), we limited the number of different orientation change amounts to five (*Figure 1B*) and focused our uncued trials on the median orientation change. Our modeled dataset is critical to addressing our general decoder hypothesis as it allows us to step beyond the restraints of physiological data to model multiple attentional modulation levels for the full range of stimulus orientations. While the main purpose of the electrophysiological data was to analyze the monkey's decoder, which could only be determined using the neuronal responses recorded from a behaving animal, the purpose of our modeled data is to compare the monkey's decoder to an ideal general decoder, which can only be determined here using a model.

Our circuit model is an extension of our previously published excitatory/inhibitory cortical network model of attention (*Huang et al., 2019*). We improved this model to allow us to calculate the network's responses to the full range of stimulus orientations by extending the three-layer model of V1 and V4 neuronal populations (*Huang et al., 2019*; *Huang et al., 2020*) to mimic realistic orientation tuning and organization in the V1 layer (*Figure 2A*).

This model is key to this initial test of the general decoder hypothesis because it allowed us to test an ideal general decoder that used the same set of neuronal weights to estimate the full range of orientations (see Materials and methods). Further, this specific model is key to our study because it is the only model (to our knowledge) that captures the effects of attention on correlated variability that have been frequently observed in electrophysiological data (*Huang et al., 2019*).

As the basis for our modeled general decoder, we first mapped the *n*-dimensional neuronal activity of our model in response to the full range of orientations to a two-dimensional space. Because the neurons were tuned for orientation, we could map the *n*-dimensional population responses to a ring

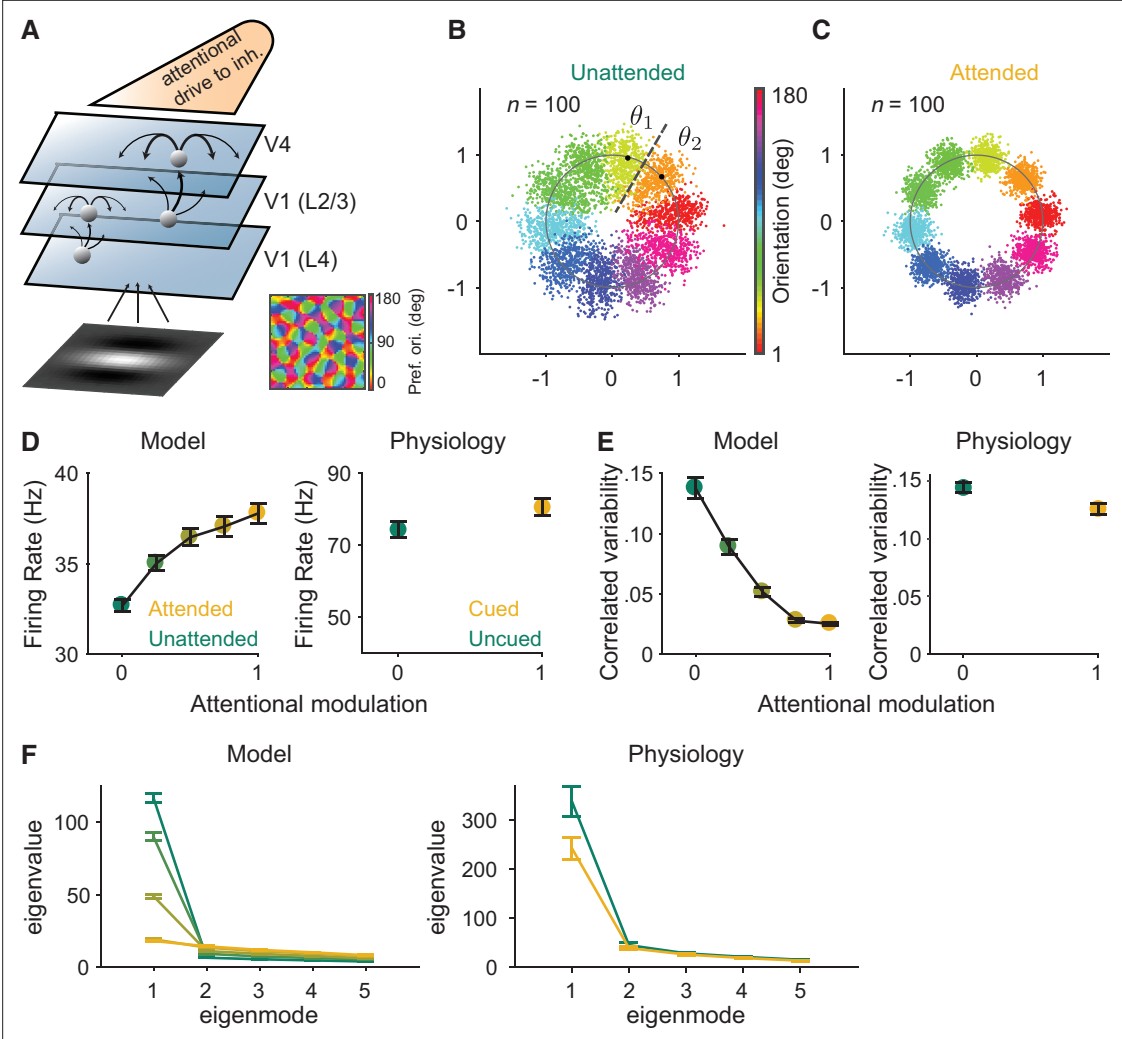

**Figure 2.** Mechanistic circuit model of attention effects. (**A**) Schematic of an excitatory and inhibitory neuronal network model of attention (*Huang et al., 2019*) that extends the three-layer, spatially ordered network to include the orientation tuning and organization of V1. The network models the hierarchical connectivity between layer 4 of V1, layers 2 and 3 of V1, and V4. In this model, attention depolarizes the inhibitory neurons in V4 and increases the feedforward projection strength from layers 2 and 3 of V1 to V4. (**B, C**) We mapped the *n*-dimensional neuronal activity of our model to a two-dimensional space (a ring). Each dot represents the neuronal activity of the simulated population on a single trial and each color represents the trials for a given orientation. These fluctuations are more elongated in the (**B**) unattended state than in the (**C**) attended state. We then calculated the effects of these attentional changes on the performance of specific and general decoders (see Materials and methods). The axes are arbitrary units. (**D–F**) Comparisons of the modeled versus electrophysiologically recorded effects of attention on V4 population activity. (**D**) Firing rates of excitatory neurons increased, (**E**) mean correlated variability decreased, and (**F**) as illustrated with the first five largest eigenvalues of the shared component of the spike count covariance matrix from the V4 neurons, attention largely reduced the eigenvalue of the first mode. Attentional state denoted by marker color for the model (yellow: most attended; green: least attended) and electrophysiological data (yellow: cued; green: uncued). For the model: 30 samplings of *n*=50 neurons. Monkey 1 data illustrated for the electrophysiological data: *n*=46 days of recorded data. SEM error bars. Also see *Figure 2—figure supplement 1*.

The online version of this article includes the following figure supplement(s) for figure 2:

**Figure supplement 1.** The model reproduces the relationship between noise and signal correlations that is key to the general decoder hypothesis.

(*Figure 2B, C*). The orientation of correlations (the shape of each color cloud in *Figure 2B*) was not an assumed parameter, and illustrates the outcome of the correlation structure and dimensionality modeled by our data. In *Figure 2B*, we can see that the fluctuations along the radial directions are much larger than those along other directions for a given orientation. This is consistent with the low-dimensional structure of the modeled neuronal activity. In our model, the fluctuations of the neurons,

mapped to the radial direction on the ring, were more elongated in the unattended state (*Figure 2B*) than in the attended state (*Figure 2C*).

Importantly, this model reproduces the correlation between noise and signal correlations (*Figure 2—figure supplement 1*) observed in electrophysiological data (*Cohen and Maunsell, 2009*; *Cohen and Kohn, 2011*). This correlation between the shared noise and the shared tuning is a key component of the general decoder hypothesis. We observed this strong relationship between noise and signal correlations in our recorded neurons (*Figure 2—figure supplement 1A*) as well as in our modeled data (*Figure 2—figure supplement 1B*). Using this model, we were able to measure the relationship between noise and signal correlations for varying strengths of attentional modulation. Consistent with the predictions of the general decoder hypothesis, attention weakened the relationship between noise and signal correlations (*Figure 2—figure supplement 1C*).

Next, we calculated the effect of this attentional change on the performances of modeled ideal specific and general decoders, which we will compare to our electrophysiological results in the next section. The specific decoder used optimal neuronal weights (for $n$ neurons in the population) based on the $n$-dimensional discrimination of two orientations ($\theta_1$ and $\theta_2$ in *Figure 2B*). The general decoder was also tested on the discrimination of those same two orientations ($\theta_1$ and $\theta_2$ in *Figure 2B*), but the neuronal weights of the general decoder were based on the neuronal population responses to all of the orientations in the ring (see Materials and methods for details). Finally, the model more than captured our electrophysiologically recorded attentional changes in V4 firing rates (*Figure 2D*), mean correlated variability (*Figure 2E*), and covariance eigenspectrum (*Figure 2F*).

## A general decoding strategy may clarify the role of mean correlated variability

First, we analyzed whether the relationship between attention, behavior, and mean correlated variability could be explained by a specific decoding strategy. This is an important first step because theoretical studies, which do not predict a relationship between mean correlated variability and performance, typically model decision-making as based on an optimal decoding strategy that maximizes the sensory information extracted from the neuronal population activity (*Kafashan et al., 2021*; *Kanitscheider et al., 2015b*; *Moreno-Bote et al., 2014*; *Pitkow et al., 2015*; *Rumyantsev et al., 2020*). A resolution to the apparent conflict between theoretical predictions and empirical results would be that perceptual performance is not based on a specific decoding strategy.

Indeed, we found that the effects of attention on the performance of the monkey's decoder did not match the effects of attention on the performance of the specific decoder (*Figure 3A*). Manipulating attention affected the performance of each decoder differently: the performance of the specific decoder was little affected by attention, while that of the monkey's decoder was strongly affected by attention.

Second, we used our circuit model of attention to test whether a modeled ideal general decoder was a better match to the physiological monkey's decoder than a modeled ideal specific decoder. The model allowed us to generate a large dataset with an experimentally unfeasible number of trials per stimulus orientation for a full ring of stimulus orientations, in multiple attention conditions (*Figure 2B and C*).

We found that the large effects of attention on the physiological monkey's decoder (*Figure 3A*) were better matched by the similarly large effects of attention on the modeled general decoder (*Figure 3B*) than by the small effects of attention on the modeled specific decoder (*Figure 3B*). In other words, the monkey's decoding strategy was most qualitatively matched to the modeled general decoder.

We note that the effects of attention on the modeled general decoder (*Figure 3B*) matched the effects of attention on the physiological monkey's decoder (*Figure 3A*) in the neuronal population size range recorded in the physiological data. But we also observed that the difference in the attentional effects on the two modeled decoders decreased with larger population sizes (right plot of *Figure 3B*). This is not noteworthy in and of itself because the attentional effects necessarily decreased as the decoders reached their information saturation points, as defined by the parameters of the modeled neurons. What we do want to note, however, is that the large attentional effects on the physiological monkey's decoders suggest that the monkeys were not working in the optimal regime near the

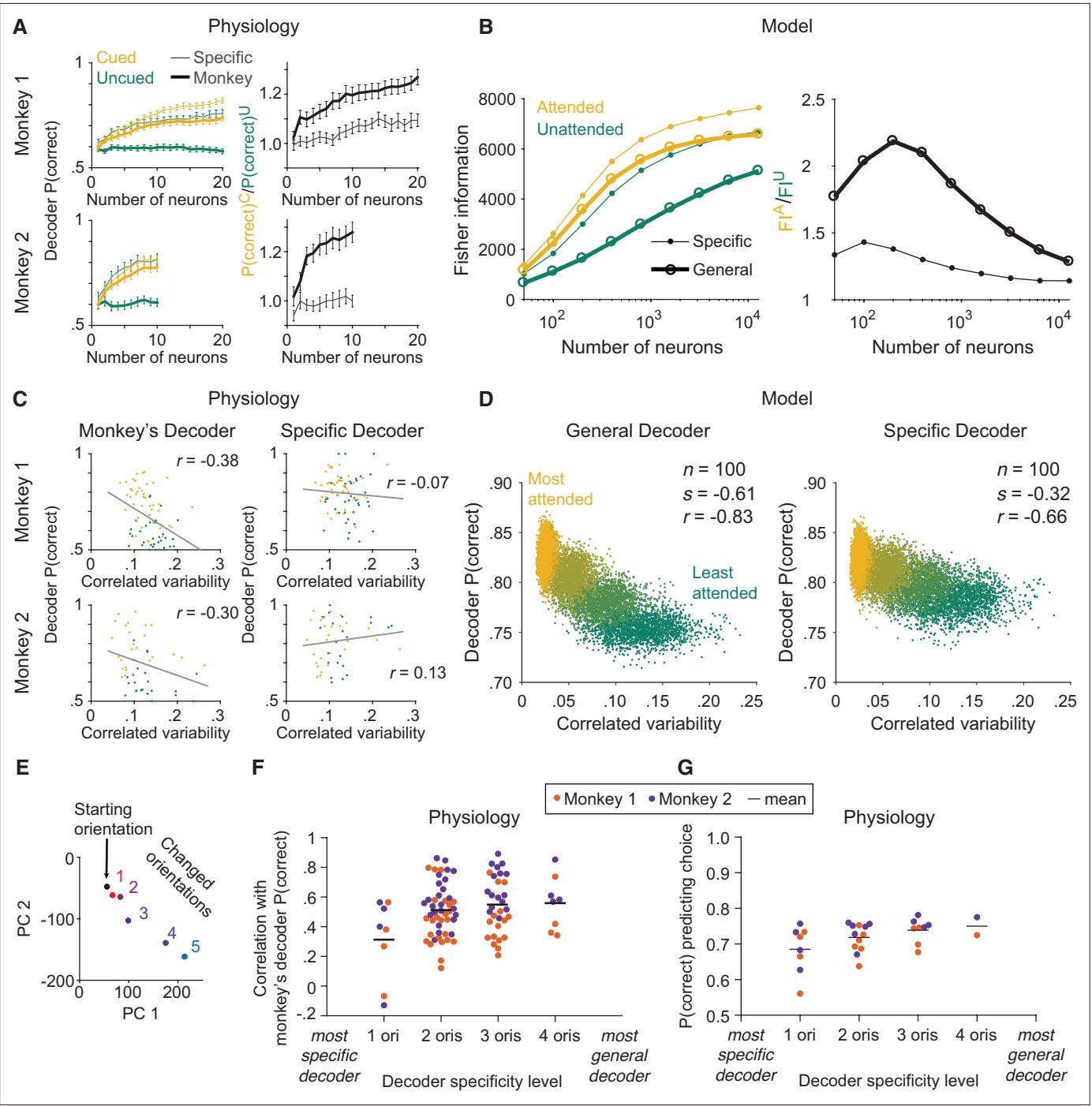

**Figure 3.** The monkey's decoding strategy was most closely matched by a general decoding strategy. (**A**) Physiological data for Monkey 1 and Monkey 2: the effect of attention on decoder performance was larger for the monkey's decoder than for the specific decoder. Left plots: decoder performance (*y*-axis; leave-one-out cross-validated proportion of trials in which the orientation was correctly identified: starting versus median changed orientation) for each neuronal population size (*x*-axis) is plotted for the specific (thin lines) and monkey's (thick lines) decoders in the cued (yellow) and uncued (green) attention conditions. Right plots: the ratio of the decoder performance in the cued versus uncued conditions is plotted for each neuronal population size. SEM error bars (Monkey 1: *n*=46 days; Monkey 2: *n*=28 days). (**B**) Modeled data: the effect of attention on decoder performance was larger for the general decoder than for the specific decoder. Left plot: the inverse of the variance of the estimation of theta (*y*-axis; equivalent to linear Fisher information for the specific decoder) for each neuronal population size (*x*-axis) is plotted for the specific decoder (small markers; Equation 1, see Materials and methods) and for the general decoder (large markers; Equation 3, see Materials and methods) in the attended (yellow) and unattended (green) conditions. Right plot: the ratio of Fisher information in the attended versus unattended conditions is plotted for each neuronal population size. (**C**) Physiological data for Monkey 1 and Monkey 2: the performance of the monkey's decoder was more related to mean correlated variability (left plots,

*Figure 3 continued*

gray lines of best fit; Monkey 1 correlation coefficient: $n$=86, or 44 days with two attention conditions plotted per day and two data points excluded – see Materials and methods, $r$=–0.38, p=5.9 × 10$^{-4}$; Monkey 2: $n$=54, or 27 days with two attention conditions plotted per day, $r$=–0.30, p=0.03) than the performance of the specific decoder (right plots; Monkey 1 correlation coefficient: $r$=–0.07, p=0.53; Monkey 2: $r$=0.13, p=0.36). For both monkeys, the correlation coefficients associated with the two decoders were significantly different from each other (Williams' procedure; Monkey 1: $t$=3.7, p=2.3 × 10$^{-4}$; Monkey 2: $t$=3.2, p=1.4 × 10$^{-3}$). Also see ***Figure 3—figure supplement 1***. (**D**) Modeled data: the performance of the general decoder was more related to mean correlated variability (left plot) than the performance of the specific decoder (right plot; number of neurons fixed at 100 and attentional state denoted by marker color, yellow to green: most attended to least attended). (**E**) An example plot of the first versus second principal component (PC) of the V4 population responses to each of the six orientations presented in the session, to justify a linear decoding strategy for the more-general decoders (starting orientation illustrated in black, five changed orientations illustrated with a red-blue color gradient from smallest to largest). Though the brain may use nonlinear decoding methods, the neuronal population representations of the small range of orientations tested per day were reasonably approximated by a line; thus, linear methods were sufficient to capture decoder performance for the physiological dataset. (**F**) Physiological data for Monkey 1 (orange) and Monkey 2 (purple): the more general the decoder ($x$-axis; number of orientation changes used to determine the decoder weights, with the decoder that best differentiated the V4 responses to the starting orientation from those to one changed orientation on the far left, and the decoder that best differentiated V4 responses to the starting orientation from those to four different changed orientations on the far right), the more correlated its performance to the performance of the monkey's decoder ($y$-axis; the across-days correlation between the performance of the monkey's decoder and the performance of the decoder specified by the $x$-axis). Mean across all points in a column illustrated by a black horizontal line (see Materials and methods for $n$ values). There was a significant correlation between decoder specificity level ($x$-axis) and the correlation with the performance of the monkey's decoder ($y$-axis; correlation coefficient: $r$=0.25, p=0.016). (**G**) The more general the decoder ($x$-axis), the better its performance predicting the monkey's choices on the median changed orientation trials ($y$-axis; the proportion of leave-one-out trials in which the decoder correctly predicted the monkey's decision as to whether the orientation was the starting orientation or the median changed orientation). Conventions as in (**F**) (see Materials and methods for $n$ values). There was a significant correlation between decoder specificity level ($x$-axis) and performance predicting the monkey's choices ($y$-axis; correlation coefficient: $r$=0.44, p=0.016).

The online version of this article includes the following figure supplement(s) for figure 3:

**Figure supplement 1.** Based on the electrophysiological data, the performance of the monkey's decoder was more related to mean correlated variability than the performance of the specific decoder within each attention condition.

**Figure supplement 2.** After shuffling trials, the performances of the modeled general and specific decoders and their relationships to the amount of correlated variability removed by the shuffling become indistinguishable.

saturation point. Instead, the monkeys appear to have been working at an inner regime that allowed them to demonstrate large effects of attention on their performance.

Third, we tested the crux of our hypothesis: that a general decoding strategy underlies the frequently reported relationship between perceptual performance and mean correlated variability. This was our most critical test toward understanding the much debated role of mean correlated variability. It would be difficult to interpret a relationship between the performance of a decoder and mean correlated variability in isolation, because a correlation between these two factors could come about through an indirect relationship. But here, we tested the explicit hypothesis that the modeled general decoder would be more related to mean correlated variability than the modeled specific decoder, just as the physiological monkey's decoder would be more strongly related to mean correlated variability than the physiological specific decoder. Such a finding would indicate that a more-general decoding strategy better explained the relationship between behavior and mean correlated variability than a more-specific decoding strategy.

Indeed, we found that just as the performance of the physiological monkey's decoder was more strongly related to mean correlated variability than the performance of the physiological specific decoder (***Figure 3C***; see ***Figure 3—figure supplement 1*** for analyses per attention condition), the performance of the modeled general decoder was more strongly related to mean correlated variability than the performance of the modeled specific decoder (***Figure 3D***; see ***Figure 3—figure supplement 2*** for trial-shuffled analyses). We modeled much stronger relationships to correlated variability (***Figure 3D***) than observed with our physiological data (***Figure 3C***). We observed that the correlation with specific decoder performance was significant with the modeled data but not with the physiological data. This is not surprising as we saw attentional effects, albeit small ones, on specific decoder performance with both the physiological and the modeled data (***Figure 3A, B***). Even small attentional effects would result in a correlation between decoder performance and mean correlated variability with a large enough range of mean correlated variability values. It is possible that with enough electrophysiological data, the performance of the specific decoder would be significantly related to correlated variability, as well. As described above, our focus is not on whether the performance of

any one decoder is significantly correlated with mean correlated variability, but on which decoder provides a better explanation of the frequently observed relationship between performance and mean correlated variability. The performance of the general decoder was more strongly related to mean correlated variability than the performance of the specific decoder.

Finally, while the circuit model allowed us to analyze an ideal general decoder for the full range of stimulus orientations in multiple attention conditions (*Figure 2B, C*; a full ring of orientations in both unattended and attended conditions), as a sanity check, we used the physiological data from the cued attention condition only (a limited set of five orientation change amounts as illustrated in *Figure 1B*, in the cued condition only as the uncued condition focused on the median orientation change amount only) to test whether more-general decoders were more related to the monkey's decoder than more-specific decoders. Would a decoder based on more stimulus orientations be more related to the monkey's decoder than a decoder based on fewer stimulus orientations?

For the physiological data, we again used linear decoders as illustrated in *Figure 1D and E*, as the V4 neuronal population representations of the limited range of stimulus orientations tested in our behavioral task (*Figure 1B*) were reasonably approximated by a line (*Figure 3E*; while a complex linear estimator was required for the modeled data based on the full range of orientations, linear methods were sufficient to capture decoder performance for the physiological dataset). First, for each monkey, we calculated the across-days correlation between the performance of the monkey's decoder and the performance of a decoder based on a single orientation change amount (*Figure 3F*, '1 ori'; see Materials and methods for more details). Next, we calculated the across-days correlation between the performance of the monkey's decoder and the performance of decoders based on two, three, or four orientation change amounts (*Figure 3F*, '2 oris' through '4 oris'). We found that the more general the decoder, the more its performance was correlated with that of the monkey's decoder. Further, the more general the decoder, the better it predicted the monkey's trial-by-trial choices on the median changed orientation trials (*Figure 3G*).

## Discussion

Our results suggest that the relationship between behavior and mean correlated variability is more consistent with observers using a more-general strategy that employs the same neuronal weights for decoding any stimulus change. The modeled general decoder better matched the attentional effects on the electrophysiological monkey's decoder and, most importantly, was more strongly related to mean correlated variability than the specific decoder. Further, based on our electrophysiological data, the more general the decoder (the more orientation change amounts used to determine the decoder weights), the more its performance was correlated with that of the monkey's decoder. Together, these results support the hypothesis that observers use a more-general decoding strategy in scenarios that require flexibility to changing stimulus conditions.

Our study also demonstrates the utility of combining electrophysiological and circuit modeling approaches to studying neural coding. Our model mimicked the correlated variability and effects of attention in our physiological data. Critically, our model produced neuronal tuning and covariance based on the constraints of an actual network capable of processing images as inputs. Using a circuit model allowed us to test a full range of stimulus orientations in multiple attention conditions, allowing us to test the effects of attention on a true general decoder for orientation.

### A fixed readout mechanism

A prior study from our lab found that attention, rather than changing the neuronal weights of the observer's decoder, reshaped neuronal population activity to better align with a fixed readout mechanism (*Ruff and Cohen, 2019*). To test whether the neuronal weights of the monkey's decoder changed across attention conditions (attended versus unattended), Ruff and Cohen switched the neuronal weights across conditions, testing the stimulus information in one attention condition with the neuronal weights from the other. They found that even with the switched weights, the performance of the monkey's decoder was still higher in the attended condition. The results of this study support the conclusion that attention reshapes neuronal activity so that a fixed readout mechanism can better read out stimulus information. In other words, differences in the performance of the monkey's decoder

across attention conditions may be due to differences in how well the neuronal activity aligns with a fixed decoder.

Our study extends the findings of Ruff and Cohen to test whether that fixed readout mechanism is determined by a general decoding strategy. Our findings support the hypothesis that observers use a general decoding strategy in the face of changing stimulus and task conditions. Our findings do not exclude other potential explanations for the suboptimality of the monkey's decoder, nor do they exclude the possibility that attention modulates decoder neuronal weights. However, our findings together with those of Ruff and Cohen shed light on why neuronal decoders are suboptimal in a manner that aligns the fixed decoder axis with the correlated variability axis (*Ni et al., 2018*; *Ruff et al., 2018*).

## A general decoding strategy in the face of unpredictable stimuli

We performed this initial test of the overarching general decoder hypothesis in the context of a change detection task along a single stimulus dimension because this type of task was used in many of the studies that reported a relationship between perceptual performance and mean correlated variability (*Cohen and Maunsell, 2009*; *Cohen and Maunsell, 2011*; *Herrero et al., 2013*; *Luo and Maunsell, 2015*; *Mayo and Maunsell, 2016*; *Nandy et al., 2017*; *Ni et al., 2018*; *Ruff and Cohen, 2016*; *Ruff and Cohen, 2019*; *Verhoef and Maunsell, 2017*; *Yan et al., 2014*; *Zénon and Krauzlis, 2012*). This simple and well-studied task provided an ideal initial test of our general decoder hypothesis.

This initial test of the general decoder hypothesis suggests that a more-general decoding strategy may explain observations in studies that use a variety of behavioral and stimulus conditions. Studies using a variety of tasks have also demonstrated a relationship between perceptual performance and mean correlated variability. These tasks include heading (*Gu et al., 2011*), orientation (*Gregoriou et al., 2014*), and contrast (*Ruff and Cohen, 2014a*; *Ruff and Cohen, 2014b*) discrimination tasks, in which the observer must respond to certain stimulus values or compare stimulus values. Some studies of discrimination tasks suggest that the relationship between perceptual performance and mean correlated variability cannot be explained by a specific decoding strategy that maximizes the amount of sensory information extracted for the task (*Clery et al., 2017*; *Gu et al., 2011*). It will be interesting to determine whether general decoders for linearly varying stimulus dimensions such as contrast (or speed, direction of motion, etc.) also provide a better explanation of the relationship between behavior and mean correlated variability than specific decoders.

On the other hand, other studies of perceptual performance have found that observers can achieve high levels of perceptual precision under certain circumstances (*Burgess et al., 1981*; *Kersten, 1987*). Such studies suggest that decoding strategies that maximize the amount of extracted sensory information might be used in certain situations. Further tests of decoding strategies in a variety of stimulus conditions and behavioral contexts will be necessary to determine when sensory information decoding prioritizes accuracy and when decoding prioritizes flexibility, or generality, over accuracy. Of particular interest is the potential role of perceptual learning (*Seitz and Watanabe, 2005*) in determining the extent to which decoding is specific (*Gu et al., 2011*; *Jeanne et al., 2013*; *Ni et al., 2018*).

## General decoders of all features would be inextricably linked to mean correlated variability

Our results address a paradox in the literature. Electrophysiological and theoretical evidence supports that there is a relationship between mean correlated variability and perceptual performance (*Abbott and Dayan, 1999*; *Clery et al., 2017*; *Haefner et al., 2013*; *Jin et al., 2019*; *Ni et al., 2018*; *Ruff and Cohen, 2019*; reviewed by *Ruff et al., 2018*). Yet, a specific decoding strategy in which different sets of neuronal weights are used to decode different stimulus changes cannot easily explain this relationship (*Kafashan et al., 2021*; *Kanitscheider et al., 2015b*; *Moreno-Bote et al., 2014*; *Pitkow et al., 2015*; *Rumyantsev et al., 2020*; reviewed by *Kohn et al., 2016*). This is because specific decoders of neuronal population activity can easily ignore changes in mean correlated noise (*Moreno-Bote et al., 2014*).

The general decoder hypothesis offers a resolution to this paradox. As a decoder becomes more general to more stimulus features, its weights will have to depend on more and more tuning properties of the neuronal population. A fully general decoder of stimuli that vary along many feature dimensions would be one whose neuronal weights depend on the tuning properties of the neurons to

all stimulus features to which they are selective. For example, two V4 neurons may both prefer vertical orientations. If they also share a color tuning preference for red, a large response from both neurons might indicate vertical orientation, the color red, or a combination of both features. A fully general decoder would need to resolve this discrepancy by choosing weights for these and other neurons that take not only their tuning for orientation but also their tuning for color into account.

Therefore, the weights of a fully general decoder would depend on the tuning of all neurons to all of the stimulus features to which they are selective. A large number of studies have shown that mean correlated variability depends on tuning similarity for all stimulus features (for review, see *Cohen and Kohn, 2011*; *Ruff et al., 2018*). The implication is that the decoding weights for a fully general decoder would depend on exactly the same properties as mean correlated variability.

This initial study of the general decoder hypothesis tested this idea in the context of a visual environment in which stimulus values only changed along a single dimension. However, our overarching hypothesis is that observers use a general decoding strategy in the complex and feature-rich visual scenes encountered in natural environments. In everyday environments, visual stimuli can change rapidly and unpredictably along many stimulus dimensions. The hypothesis that such a truly general decoder explains the relationship between perceptual performance and mean correlated variability is suggested by our finding that the modeled general decoder for orientation was more strongly related to mean correlated variability than the modeled specific decoder (*Figure 3D*). Future tests of a general decoder for multiple stimulus features would be needed to determine if this decoding strategy is used in the face of multiple changing stimulus features. Further, such tests would need to consider alternative hypotheses for how sensory information is decoded when observing multiple aspects of a stimulus (*Berkes et al., 2009*; *Deneve, 2012*; *Lorteije et al., 2015*). Studies that use complex or naturalistic visual stimuli may be ideal for further investigations of this hypothesis.

The purpose of this study was to investigate the relationship between mean correlated variability and a general decoder. We made an initial test of the overarching hypothesis that observers use a general decoding strategy in feature-rich environments by testing whether a decoder optimized for a broader range of stimulus values better matched the decoder actually used by the monkeys than a specific decoder optimized for a narrower range of stimulus values. We purposefully did not make claims about the utility of correlated variability relative to hypothetical situations in which correlated variability does not exist in the responses of a group of neurons, as we suspect that this is not a physiologically realistic condition. Studies that causally manipulate the level of correlated variability in neuronal populations to measure the true physiological and behavioral effects of increasing or decreasing correlated variability levels, through pharmacological or genetic means, may provide important insights into the impact of correlated variability on various decoding strategies.

In our model, which was designed to mimic real data, attention changed many aspects of neural responses besides just correlated variability. It is therefore possible that any relationship between decoding performance and correlated variability is mostly caused by those concomitant changes. Thus, we used the many trials in our modeled data to test the effects of randomly shuffling the trial order per modeled neuron. These shuffled data resulted in the modeled general and specific decoders becoming essentially indistinguishable in their relationships with the removed correlated variability (*Figure 3—figure supplement 2*), with those removed correlations essentially representing attention condition. The effects of attention on many aspects of neuronal population activity have been well documented, including effects on neuronal firing rates and on both individual and shared trial-to-trial response variability (*Cohen and Maunsell, 2009*; *Cohen and Maunsell, 2011*; *Herrero et al., 2013*; *Luo and Maunsell, 2015*; *Mayo and Maunsell, 2016*; *Mitchell et al., 2009*; *Nandy et al., 2017*; *Ni et al., 2018*; *Ruff and Cohen, 2014a*; *Ruff and Cohen, 2014b*; *Ruff and Cohen, 2016*; *Ruff and Cohen, 2019*; *Zénon and Krauzlis, 2012*). The simulated neurons in our model captured many of these attention effects (*Figure 2D–F*; *Huang et al., 2019*). Much theoretical work has supported that in the presence of correlations between neuronal responses, specifically differential or information-limiting correlations, limits on decoding performance will be dominated by these correlations (*Moreno-Bote et al., 2014*; for review, see *Kohn et al., 2016*). Removing these correlations by shuffling the trials results in many changes; in particular, our trial-shuffled data demonstrate the well-documented linear growth in Fisher information that is expected with increasing numbers of neurons (*Figure 3—figure supplement 2A–C*; *Averbeck et al., 2006*; *Kohn et al., 2016*; *Shadlen et al., 1996*). Our trial-shuffled analysis illustrates that removing correlations results in decoder performance being dominated by

other effects of attention on neuronal activity, such as the firing rates (gains) of the neurons. Our model reproduces the gain effects of attention on neuronal firing rates observed in electrophysiological data (*Figure 2D*) which, in the absence of correlations, increases the sensitivity of the population (*Averbeck et al., 2006*; *Kohn et al., 2016*; *Shadlen et al., 1996*). In summary, general and specific decoder performances had indistinguishable relationships with the amount of correlated variability removed by the trial shuffling (*Figure 3—figure supplement 2D, E*), suggesting that decoder performance became dominated by attention-related firing rate gains intrinsic to our model.

In conclusion, the findings of this study support the usefulness of a framework that relates sensory information decoding to behavior (for review, see *Panzeri et al., 2017*). By first determining the decoder that guided each monkey's behavioral choices, we were able to compare the monkey's decoder to modeled specific and general decoders to test our hypothesis. These results demonstrate that constraining analyses of neuronal data by behavior can provide important insights into the neurobiological mechanisms underlying perception and cognition.

## Materials and methods

### Electrophysiological recordings

The subjects were two adult male rhesus monkeys (*Macaca mulatta*, 8 and 10 kg). All animal procedures were approved by the Institutional Animal Care and Use Committees of the University of Pittsburgh and Carnegie Mellon University (Protocol #17071123). Different aspects of these data were presented previously (*Ni et al., 2018*). We recorded extracellularly from single units and sorted multiunit clusters (the term 'unit' refers to either; see *Ni et al., 2018*) in V4 of the left hemisphere using chronically implanted 96-channel microelectrode arrays (Blackrock Microsystems) with 1 mm long electrodes. We performed all spiking sorting manually using Plexon's Offline Sorter (version 3.3.5, Plexon).

We only included a recorded unit if its stimulus-driven firing rate was both greater than 10 Hz and significantly higher than the baseline firing rate (baseline calculated as the firing rate in the 100 ms window immediately prior to the onset of the first stimulus per trial; two-sided Wilcoxon signed rank test: $p<10^{-10}$). The population size of simultaneously recorded units was 8–45 units (mean 39) per day for Monkey 1 and 7–31 units (mean 19) per day for Monkey 2.

No statistical methods were used to predetermine our sample sizes of subjects or recorded units, but our sample sizes are similar to those used in previous publications that analyzed neuronal and behavioral data similar to the data analyzed here (*Cohen and Maunsell, 2009*; *Ni et al., 2018*).

### Behavioral task

The monkeys performed a change detection task (*Figure 1A*; *Cohen and Maunsell, 2009*) with multiple orientation change options (*Figure 1B*) and cued attention (*Posner, 1980*) while we recorded electrophysiological data. We presented visual stimuli on a CRT monitor (calibrated to linearize intensity; 1024×768 pixels; 120 Hz refresh rate) placed 52 cm from the monkey, using custom software written in MATLAB (Psychophysics Toolbox; *Brainard, 1997*; *Pelli, 1997*). We monitored each monkey's eye position using an infrared eye tracker (Eyelink 1000; SR Research) and recorded eye position, neuronal responses (30,000 samples/s), and the signal from a photodiode to align neuronal responses to stimulus presentation times (30,000 samples/s) using Ripple hardware.

A trial began when a monkey fixed its gaze on a small, central spot on the video display while two peripheral Gabor stimuli (one overlapping the RFs of the recorded neurons, the other in the opposite visual hemifield; *Figure 1C*) synchronously flashed on (for 200 ms) and off (for a randomized period between 200 and 400 ms) at the same starting orientation until at a random, unsignaled time the orientation of one of the stimuli changed. The monkey received a liquid reward for making a saccade to the changed stimulus within 400 ms of its onset.

Attention was cued in blocks of trials, with each block preceded by 10 instruction trials that cued one of the two stimulus locations by only presenting stimuli at that location. Each block consisted of approximately 125 orientation change trials. In each block, the orientation change occurred at the cued location in 80% of the change trials and at the uncued location in 20% of the change trials. Catch trials were intermixed, in which no orientation change occurred within the maximum of 12 stimulus presentations. In catch trials, the monkeys were rewarded for maintaining fixation. Blocks of trials

with attention cued to the left hemifield location or to the right hemifield location were presented in alternating order within a recording day.

The changed orientation at the cued location was randomly selected per trial from one of five changed orientations (with the constraint of required numbers of presentations per changed orientation per block; *Figure 1B*) such that the monkeys could not predict which orientation change amount was to be detected on any given trial. The changed orientation at the uncued location was randomly either the median (20 trials per block) or the largest orientation change amount (5 trials per block). Uncued changes were collected mainly for the median change amount to maximize the number of uncued trials collected for one change amount. All analyses of the effects of attention analyzed the cued versus uncued median change amounts.

The size, location, and spatial frequency of the Gabor stimuli were fixed across all recording days. These three parameters were determined in advance of recording the data presented here, using a receptive field mapping task. These parameters were set to maximize the neuronal responses recorded by the array.

The starting orientation (*Figure 1A, B*) was identical for all trials within a day. We changed the starting orientation by 15° for each new day of recording. We also changed the five changed orientation options (*Figure 1B*) for each new day of recording, to maintain the task at approximately the same level of difficulty across days. The five changed orientation options were always the same within one session of trials, with one session equaling two blocks of trials: one block of trials with the left stimulus cued, and one block of trials with the right stimulus cued (*Figure 1C*). We sometimes changed the five orientation options between sessions within a day, again to maintain a consistent level of task difficulty. For those days, we binned the orientation change amounts into five bins based on their log distribution.

## Electrophysiological data analysis

The data presented are from 46 days of recording for Monkey 1 and 28 days of recording for Monkey 2. Instruction trials were not included in any analyses. Only trials in which the orientation changes occurred at the RF location (*Figure 1C*) and catch trials were analyzed (see below for specific inclusions per analysis). The first stimulus presentation of each trial was excluded from all analyses to minimize temporal non-stationarities due to adaptation.

Firing rates (*Figure 2D*), mean correlated variability (*Figures 2E and 3C*), and covariance eigenspectrum analyses (*Figure 2F*) were calculated based on orientation change trials on which the monkey correctly detected the change as well as on catch trials. From these trials, only the starting orientation stimulus presentations were included in the analyses. The firing rate per stimulus presentation was based on the spike count response between 60 and 260 ms after stimulus onset to account for V4 latency. These analyses were performed per recording day (such that all starting orientation stimuli analyzed together were identical). Data were presented as the mean per day (*Figure 3C*) or across days (*Figure 2D–F*) per attention condition (cued or uncued).

We defined the mean correlated variability of each pair of simultaneously recorded units (quantified as the noise correlation or spike count correlation; *Cohen and Kohn, 2011*) as the correlation between the firing rates of the two units in response to repeated presentations of the same stimulus (the starting orientation). This measure of mean correlated variability represents correlations in noise rather than in signal because the visual stimulus was always the same.

For *Figure 3C*, we used Williams' procedure for comparing correlated correlation coefficients (*Howell, 2007*) to compare the across-days correlation between the performance of the monkey's decoder and the mean correlated variability of the V4 population to the across-days correlation between the performance of the specific decoder and the mean correlated variability of the V4 population.

For Monkey 1, two outlier points (uncued trials for each of 2 days) with mean correlated variability values greater than 0.35 were excluded from analysis based on the Tukey method (see *Figure 3C* for the range of included correlated variability values for Monkey 1). For *Figure 3C*, with the excluded points included, the correlation coefficients were qualitatively unchanged: for the monkey's decoder, $n=88$, or 44 days (see below for data included in decoder analyses) with two attention conditions plotted per day, $r=-0.34$, $p=1.7 \times 10^{-3}$; for the specific decoder, $r=-0.22$, $p=0.05$.

## V4 population specific decoder

For *Figure 3A and C*, we calculated the performance of a specific decoder based on the electrophysiologically recorded V4 neuronal population data (*Ni et al., 2018*). To avoid artifacts in neuronal firing rates due to eye movements in response to the changed orientation, all V4 population decoder analyses were based on neuronal firing rates during an abbreviated time window: 60–130 ms after stimulus onset.

The specific decoder was a linear classifier trained to best differentiate the V4 population responses to the median changed orientation from the V4 responses to the starting orientation presented immediately before it (*Figure 1D*; first and second principal components shown for illustrative purposes only – analyses were based on neuronal population firing rates). The neuronal weights were calculated per day and per attention condition.

Decoder performance was quantified as the leave-one-out cross-validated proportion of correctly identified orientations (median changed orientation or starting orientation). For *Figure 3A*, decoder performance was analyzed per number of neurons (*x*-axis). Per neuronal population size, the most responsive neurons (ranked by evoked response: stimulus-evoked firing rate minus baseline firing rate) were analyzed.

For *Figure 3C*, decoder performance was illustrated for a set number of neurons (Monkey 1: 20 units, Monkey 2: 10 units). The number of neurons analyzed for these plots was selected to maximize the number of included neurons and recording days (Monkey 1: *n*=44 days, 2 days with 8 and 19 recorded units excluded; Monkey 2: *n*=27 days, 1 day with 7 recorded units excluded).

## V4 population monkey's decoder

For *Figure 3A, C*, we calculated the performance of the monkey's choice decoder as well (*Figure 1E*). The monkey's decoder was a linear classifier trained on the same set of V4 population responses as the specific decoder described above (the neuronal population firing rates in response to the median changed orientation and in response to the starting orientation presented immediately before it). However, unlike the specific decoder, the monkey's decoder was trained to best differentiate the V4 population responses when the monkey made a saccade indicating it detected the orientation change from the V4 responses when the monkey did not make a saccade (both correctly in response to the starting orientation and incorrectly when the monkey missed the changed orientation).

Decoder performance was quantified just as it was for the specific decoder described above: as the leave-one-out cross-validated proportion of correctly identified orientations (median changed orientation or starting orientation). *Figure 3A, C* was calculated as described above for the specific decoder.

In summary, the physiological specific and monkey's decoders were trained on different classifications of the same set of V4 responses and thus had different neuronal weights. However, everything else about how they were analyzed was the same, including that their performance was tested on the same task of correctly identifying whether each left-out orientation was the median changed orientation or the starting orientation.

## V4 population more-general versus more-specific decoders

For *Figure 3F*, we calculated decoders that were increasingly more general, to compare their performance to that of the monkey's decoder. As described above, due to the limited number of behavioral trials collected per day, we could not calculate an ideal general decoder for orientation based on the physiological data as we could based on the modeled data. However, as a sanity check, we wanted to check whether more-general decoders were more strongly related to the monkey's decoder than more-specific decoders. Additionally, we only analyzed trials from the cued attention condition (uncued trials were mainly collected for the median orientation change amount, which is why the median orientation change trials were analyzed in the attention analyses illustrated in *Figure 3A, C*). As with *Figure 3C* as described above, decoder performance was illustrated for a set number of neurons (Monkey 1: 20 units, Monkey 2: 10 units).

As above, the monkey's decoder was trained on the median (third; *Figure 1B*) orientation change trials. The monkey's decoder was trained to best differentiate the V4 population responses when the monkey made a saccade (indicating it detected the orientation change) from the V4 responses when the monkey did not make a saccade. Importantly, we sought to avoid the relationship that would be inherent between the monkey's decoder and any decoder based on those same median (third)

orientation change trials. Thus, only the neuronal weights for the monkey's decoder were based on the median (third) orientation change trials. The weights of all of the other decoders in *Figure 3F* were based on trials other than the median orientation change trials.

The most specific decoder tested in *Figure 3F* was trained to best differentiate the V4 population responses to one changed orientation from the V4 responses to the starting orientation presented immediately before it. The performance of this specific decoder was quantified as the leave-one-out cross-validated proportion of trials on which the decoder correctly identified whether the left-out orientation was that one changed orientation or the starting orientation. The one changed orientation that was tested was either the first, second, fourth, or fifth largest changed orientation (all but the median changed orientation, which was excluded from this analysis due to the inherent relationship between any decoder that included this orientation and the monkey's decoder which was based on this orientation, as noted above). We calculated the performance of each decoder on each day of recording. We then calculated the across-days correlation between the performance of this specific decoder and the performance of the monkey's decoder (described above). The correlation coefficient was plotted in the '1 ori' column of *Figure 3F* for each monkey ($n$=8 decoders; 1 decoder for each of the 4 included changed orientations, for each of the 2 monkeys).

We performed this same procedure for increasingly more-general decoders. The decoders illustrated in the '2 oris' column were trained to best differentiate the V4 responses to two changed orientations from the V4 responses to the starting orientation. The two changed orientations used for each '2 oris' decoder were chosen from the four possibilities: the first, second, fourth, and fifth (max) largest changed orientations ($n$=48 decoders; 1 decoder for each of the 6 combinations of 2 changed orientations, tested on each of the 4 included changed orientations, for each of the 2 monkeys). We calculated the performance of each '2 oris' decoder on the same task as above: identifying whether the orientation was the changed orientation (the first, second, fourth, or fifth largest changed orientation) or the starting orientation. Again, we calculated the across-days correlation between the performance of each '2 oris' decoder and the performance of the monkey's decoder.

Each '3 oris' decoder was trained to best differentiate the V4 responses to three changed orientations from the V4 responses to the starting orientation ($n$=32 decoders; 1 decoder for each of the 4 combinations of 3 changed orientations, tested on each of the 4 included changed orientations, for each of the 2 monkeys). The '4 oris' decoder for each monkey was trained to best differentiate the V4 responses to four changed orientations from the V4 responses to the starting orientation ($n$=8 decoders; 1 decoder for the 1 combination of 4 changed orientations, tested on each of the 4 included changed orientations, for each of the 2 monkeys).

For *Figure 3G*, we performanced analyses similar to those performed for *Figure 3F*, in that we tested each stimulus decoder: '1 ori' decoders ($n$=8 decoders; 1 specific decoder for either the first, second, fourth, or fifth largest changed orientation, for each of the 2 monkeys), '2 oris' decoders ($n$=12 decoders; 1 decoder for each of the 6 combinations of 2 changed orientations, for each of the 2 monkeys), '3 oris' decoders ($n$=8 decoders; 1 decoder for each of the 4 combinations of 3 changed orientations, for each of the 2 monkeys), and '4 oris' decoders ($n$=2 decoders; 1 decoder for the 1 combination of 4 changed orientations, for each of the 2 monkeys). However, unlike in *Figure 3F*, where the performance of the stimulus decoders was compared to the performance of the monkey's decoder on the median orientation change trials, here, we calculated the performance of the stimulus decoder when tasked with predicting the trial-by-trial choices that the monkey made on the median orientation change trials. We plotted the proportion of leave-one-out trials in which each decoder correctly predicted the monkey's choice as to whether the orientation was the starting orientation or the median changed orientation.

## Network model description

The network model is similar to the one in *Huang et al., 2019*. Briefly, the network consists of three modeled stages: (1) layer (L) 4 neurons of V1, (2) L2/3 neurons of V1, and (3) L2/3 neurons of V4 (*Figure 2A*). Neurons from each area are arranged on a uniform grid covering a unit square $\Gamma = [-0.5, 0.5] \times [-0.5, 0.5]$. The L4 neurons of V1 are modeled as a population of excitatory neurons, the spikes of which are taken as inhomogeneous Poisson processes with rates determined as below. The L2/3 of V1 and V4 populations are recurrently coupled networks with excitatory and inhibitory neurons. Each neuron is modeled as an exponential integrate-and-fire (EIF) neuron. The connection

probability between neurons decays with distance. The network model captures many attention-mediated changes on neuronal responses, such as the reduction of correlated variability within each visual area, increase in correlated variability between visual areas, and the quenching of the low-dimensional correlated variability by attention. The network parameters are the same as those used in *Huang et al., 2019* except the following. The feedforward projection width from V1 (L2/3) to V4 is $\alpha_{\text{ffwd}}^{(3)} = 0.05$. The feedforward strength from V1 (L2/3) to V4 is $[J_{\text{eF}}^3, J_{\text{iF}}^3] = \gamma[1, 0.4]$. From the most unattended state to the most attended state (attentional modulation scale from 0 to 1), $\gamma$ varies from 20 to 23 mV, and the depolarizing current to the inhibitory neurons in V4, $\mu_i$, varies from 0 to 0.5 mV/ms (*Figure 2*, *Figure 3B, D*).

The model differs from the previous model (*Huang et al., 2019*) in the following ways. We modeled the V1 (L4) neurons as orientation selective filters with static nonlinearity and Poisson spike generation (*Kanitscheider et al., 2015b*). The firing rate of each neuron $i$ is $r_i(\theta, t) = [F_i \times \tilde{I}(\theta, t)]_+$, where $F_i$ is a Gabor filter and $\tilde{I}(\theta, t)$ is a Gabor image corrupted by independent noise following the Ornstein-Uhlenbeck process,

$$\tilde{I}(\theta, t) = I(\theta) + \eta(t) \quad \text{and} \quad \tau_n d\eta_i = -\eta_i dt + \sigma_n dW,$$

with $\tau_n = 40$ ms and $\sigma_n = 3.5$. The Gabor filters were normalized such that the mean firing rate of V1 (L4) neurons was 10 Hz. Spike trains of V1 (L4) neurons were generated as inhomogeneous Poisson processes with rate $r_i(\theta, t)$. The Gabor image is defined on $\Gamma$ with $25 \times 25$ pixels with spatial Gaussian envelope width $\sigma = 0.2$, spatial wavelength $\lambda = 0.6$ and phase $\phi = 0$ (*Kanitscheider et al., 2015b*, Supp Equation 6 ). The Gabor filters of V1 (L4) neurons had the same $\sigma$, $\lambda$ and $\phi$ as the image (*Kanitscheider et al., 2015b*, Supp Equation 5 ). The orientation $\theta$ was normalized between 0 and 1. The orientation preference map of L4 neurons in V1 was generated using the formula from *Kaschube et al., 2010* (Supp Equation 20) with average column spacing $\Lambda = 0.2$.

Each network simulation was 20 sec long consisting of alternating OFF (300 ms) and ON (200 ms) intervals. During OFF intervals, spike trains of Layer 1 neurons were independent Poisson processes with rate $r_X = 5$ Hz. An image with a randomly selected orientation was presented during ON intervals. Spike counts during the ON intervals were used to compute the performance of different decoders and correlated variability. The first spike count in each simulation was excluded. For each parameter condition, the connectivity matrices were fixed for all simulations. The initial states of each neuron's membrane potential were randomized in each simulation. All simulations were performed on the CNBC Cluster in the University of Pittsburgh. All simulations were written in a combination of C and Matlab (Matlab R 2015a, Mathworks). The differential equations of the neuron model were solved using the forward Euler method with time step 0.01 ms.

## Network model specific decoder

Let **r** be a vector of spike counts from all neurons on a single trial, **f** be the tuning curve function, and $\Sigma$ be the covariance matrix. Consider a fine discrimination task of two orientations $\theta^+ = \theta_0 + d\theta$ and $\theta^- = \theta_0 - d\theta$. The specific decoder is a local linear estimator:

$$\hat{\theta} = \theta_0 + \mathbf{w}^T (\mathbf{r} - \frac{\mathbf{f}(\theta^+) + \mathbf{f}(\theta^-)}{2}).$$

The optimal weight to minimize the mean squared error over all trials, $E = \langle |\hat{\theta} - \theta|^2 \rangle$, is

$$\mathbf{w}_{\text{opt}}^s = \frac{\Sigma^{-1} \mathbf{f}'}{\mathbf{f}' \Sigma^{-1} \mathbf{f}'}.$$

The linear Fisher information is equivalent to the inverse of the variance of the optimal specific decoder:

$$I = \frac{1}{\text{Var}(\hat{\theta}_{opt} | \theta^i)} = \mathbf{f}' \Sigma^{-1} \mathbf{f}'.$$

The linear Fisher information is estimated with bias-correction (*Figure 3B*; *Kanitscheider et al., 2015a*):

$$\hat{I} = \frac{(\mathbf{f}^+ - \mathbf{f}^-)^T}{d\theta} \left( \frac{\Sigma^+ + \Sigma^-}{2} \right)^{-1} \frac{(\mathbf{f}^+ - \mathbf{f}^-)}{d\theta} \left( \frac{2N_{\text{tr}} - N - 3}{2N_{\text{tr}} - 2} \right) - \frac{2N}{N_{\text{tr}} d\theta^2}, \tag{1}$$

where $\mathbf{f}^i$ and $\Sigma^i$ are the empirical mean and covariance, respectively, for $\theta^i$, $i \in \{+, -\}$. The number of neurons sampled is $N$, and the number of trials for each $\theta^i$ is $N_{\mathrm{tr}}$. In simulations, we used $\theta_0 = 0.5$ and $d\theta = 0.01$. There were 58,500 spike counts in total for $\theta^+$ and $\theta^-$.

## Network model general decoder

The general decoder is a complex linear estimator $\hat{z} = \mathbf{w^T r}$ (*Shamir and Sompolinsky, 2006*) where $\mathbf{w}$ is fixed for all $\theta$. The estimator $\hat{z}$ maps the population activity $\mathbf{r}$ in response to all orientations to a circle ($z = e^{i\theta}$ in complex domain). The estimation of orientation is $\hat{\theta} = \arg(\hat{z})$. The optimal weight $\mathbf{w}_{\mathrm{opt}}^g$ that minimizes the mean squared error, $E(\mathbf{w}) = \langle |\hat{z} - z|^2 \rangle_{\theta, \mathbf{r}}$, averaged over all $\theta$ and trials of $\mathbf{r}$, is

$$\mathbf{w}_{\mathrm{opt}}^g = \langle \Sigma(\theta) + \mathbf{ff}^T \rangle_\theta^{-1} \langle \mathbf{f} e^{i\theta} \rangle_\theta, \tag{2}$$

The mean squared error of the optimal general decoder is

$$E(\mathbf{w}_{\mathrm{opt}}^g) = 1 - (\langle \mathbf{f} e^{i\theta} \rangle_\theta)^* \langle \Sigma(\theta) + \mathbf{ff}^T \rangle_\theta^{-1} (\langle \mathbf{f} e^{i\theta} \rangle_\theta),$$

where $*$ denotes the conjugate transpose. Hence, the estimation error of $\hat{z}$ depends on both the covariance matrix, $\Sigma$, and tuning similarity, $\mathbf{ff}^T$. The performance of the general decoder is measured as $I_g = 1/\mathrm{Var}(\hat{\theta})$ (*Figure 3B*). The estimation of $I_g$ is

$$\hat{I}_g = \frac{1}{\mathrm{Var}(\arg((\mathbf{w}_{\mathrm{opt}}^g)^T \mathbf{r}) - \theta)} \frac{N_{\mathrm{tr}} - N - 2}{N_{\mathrm{tr}} - 1}, \tag{3}$$

where $N_{\mathrm{tr}}$ is the total number of trials for all $\theta$'s. In simulations, we used 50 $\theta$'s uniformly spaced between 0 and 1. There were 117,000 trials in total for all $\theta$'s.

## Dependence of network model decoders' performance on correlated variability (Figure 3D)

We trained specific and general decoders on the same spike count dataset ($\mathbf{r}$) in response to pairs of orientations, $\theta_1$ and $\theta_2$ (with difference $\Delta\theta = 0.04$). The specific decoder was trained on the $N$-dimensional space of neural responses, using a support vector machine model with two-fold cross-validation to linearly classify $\mathbf{r}$ for the two orientations. The general decoder first maps $\mathbf{r}$ to a two-dimensional plane $\hat{z} = (\mathbf{w}_{\mathrm{opt}}^g)^T \mathbf{r}$ using the optimal weight $\mathbf{w}_{\mathrm{opt}}^g$ (*Equation 2*) computed with the spike counts of all orientations. Then a two-dimensional support vector machine model with two-fold cross-validation was trained to linearly classify $\hat{z}$ for $\theta_1$ and $\theta_2$. The correlated variability was computed from the spike counts data for $\theta_1$ of each pair. There were 200 samplings of $N = 100$ excitatory neurons from the V4 network, and 10 orientation pairs varying between 0 and 1. There were on average 2,340 trials for each $\theta$.

## Factor analysis for network model

Let $x \in \mathbb{R}^{n \times 1}$ be the spike counts from $n$ simultaneously recorded neurons. Factor analysis assumes that $x$ is a multi-variable Gaussian process:

$$x \sim \mathcal{N}(\mu, LL^T + \Psi)$$

where $\mu \in \mathbb{R}^{n \times 1}$ is the mean spike counts, $L \in \mathbb{R}^{n \times m}$ is the loading matrix of the $m$ latent variables and $\Psi \in \mathbb{R}^{n \times 1}$ is a diagonal matrix of independent variances for each neuron (*Cunningham and Yu, 2014*). We chose $m = 5$ and computed the eigenvalues of $LL^T$, $\lambda_i$ ($i = 1, 2, ..., m$), ranked in descending order. Spike counts were collected using a 200 ms window. There were on average 2,340 trials per attentional condition.

## Code availability

Computer code for all simulations and analysis of the resulting data will be available at https://github.com/hcc11/GeneralDecoder; swh:1:rev:5056b409f2d943736b0478ff7ff38dd247b468b5, *Ni et al., 2022*.

# Acknowledgements

We thank Douglas A Ruff, Joshua J Alberts, and Jen Symmonds for assistance with data collection. We thank Karen McCracken for technical assistance. We thank Alexandre Pouget and Douglas A Ruff for comments on a previous version of the manuscript.

# Additional information

## Funding

| Funder | Grant reference number | Author |
|---|---|---|
| National Institutes of Health | 1K99NS118117-01 | Amy M Ni |
| Simons Foundation | | Amy M Ni Chengcheng Huang Brent Doiron Marlene R Cohen |
| Swartz Foundation | Fellowship #2017-7 | Chengcheng Huang |
| National Institutes of Health | 1U19NS107613-01 | Brent Doiron |
| National Institutes of Health | R01EB026953 | Brent Doiron |
| Vannevar Bush faculty fellowship | N00014-18-1-2002 | Brent Doiron |
| National Institutes of Health | 4R00EY020844-03 | Marlene R Cohen |
| National Institutes of Health | R01 EY022930 | Marlene R Cohen |
| National Institutes of Health | Core Grant P30 EY008098s | Marlene R Cohen |
| Whitehall Foundation | | Marlene R Cohen |
| Klingenstein-Simons Fellowship | | Marlene R Cohen |
| Sloan Research Fellowship | | Marlene R Cohen |
| McKnight Foundation | | Marlene R Cohen |

The funders had no role in study design, data collection and interpretation, or the decision to submit the work for publication.

## Author contributions

Amy M Ni, Conceptualization, Data curation, Formal analysis, Funding acquisition, Investigation, Methodology, Visualization, Writing – original draft, Writing – review and editing; Chengcheng Huang, Conceptualization, Formal analysis, Funding acquisition, Investigation, Methodology, Resources, Software, Visualization, Writing – original draft, Writing – review and editing; Brent Doiron, Marlene R Cohen, Conceptualization, Funding acquisition, Investigation, Methodology, Resources, Supervision, Writing – original draft, Writing – review and editing

## Author ORCIDs

Amy M Ni http://orcid.org/0000-0002-1746-9206
Marlene R Cohen http://orcid.org/0000-0001-8583-4300

## Ethics

All animal procedures were approved by the Institutional Animal Care and Use Committees of the University of Pittsburgh and Carnegie Mellon University (Protocol #17071123).

## Decision letter and Author response

Decision letter https://doi.org/10.7554/eLife.67258.sa1
Author response https://doi.org/10.7554/eLife.67258.sa2

# Additional files

## Supplementary files
• Transparent reporting form

## Data availability
Electrophysiological data analyzed in this manuscript are freely and publicly available at the Open Science Framework at https://doi.org/10.17605/OSF.IO/RN7TU. Computer code for all simulations and analysis of the resulting data are freely and publicly available at https://github.com/hcc11/GeneralDecoder (copy archived at swh:1:rev:5056b409f2d943736b0478ff7ff38dd247b468b5).

The following dataset was generated:

| Author(s) | Year | Dataset title | Dataset URL | Database and Identifier |
|---|---|---|---|---|
| Ni Amy | 2022 | NiHuangDoironCohen2022 | https://doi.org/10.17605/OSF.IO/RN7TU | Open Science Framework, 10.17605/OSF.IO/RN7TU |

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
