## [Editor Report]

Empirical findings have established that experimental manipulations which increase perceptual accuracy also generally reduce the amount of shared variability between neurons in the visual cortex. To explain this observation, this study combines neurophysiology data and a network model of visual cortex and tests the hypothesis that perception relies on a "general" decoding strategy. The results suggest that the brain seeks to decode arbitrary changes in stimuli that appear in the environment.

---

## [Decision Letter]

**Decision letter after peer review:**

Thank you for submitting your article "A general decoding strategy explains the relationship between behavior and correlated variability" for consideration by *eLife*. Your article has been reviewed by 3 peer reviewers, one of whom is a member of our Board of Reviewing Editors, and the evaluation has been overseen by Tirin Moore as the Senior Editor. The reviewers have opted to remain anonymous.

Essential revisions:

A highly robust result when investigating how neural population activity is impacted by performance in a task is that the trial to trial correlations (noise correlations) between neurons is reduced as performance increases. However, the theoretical and experimental literature so far has failed to account for this robust link since reduced noise correlations do not systematically contribute to improved availability or transmission of information (often measured using decoding of stimulus identity). This paper sets out to address this discrepancy by proposing that the key to linking noise correlations to decoding and thus bridging the gap with performance is to rethink the decoders we use : instead of decoders optimized to the specific task imposed on the animal on any given trial (A vs B / B vs C / A vs C), they hypothesize that we should favor a decoder optimized for a general readout of stimulus properties (A vs B vs C).

To test this hypothesis, the authors use a combination of quantitative data analysis and mechanistic network modeling. Data were recorded from neuronal populations in area V4 of two monkeys trained to perform an orientation change detection task, where the magnitude of orientation change could vary across trials, and the change could happen at cued (attended) or uncued (unattended) locations in the visual field. The model, which extends previous work by the authors, reproduces many basic features of the data, and both the model and data offer support (with one exception, details below) for the hypothesis.

The reviewers agreed that this is a potentially important contribution, that addresses a widely observed, but puzzling, relation between perceptual performance and noise correlations. The clarity of the hypothesis, and the combination of data analysis and computational modelling are two essential strengths of the paper. Nonetheless, as detailed below, the reviewers believe the manuscript clarity could be further improved in several points, and some additional analysis of the data would provide more straightforward test of the hypothesis.

Essential revisions:

1. it would be important to verify that the model reproduces the correlation between noise and signal correlations since this is really a key argument leading to the author's hypothesis. One possibility would be to make a scatterplot of these two correlations for all neurons for both neural data and the model and for example compare slopes. The slope for the model could be shown for a range of levels of attentional modulation and for the neural data in attended vs unattended as they already do in fig2d of Fig2e. The authors could provide insight into the difference between the specific and general decoder by directly assessing the alignment between the decoders and the noise dimension. This could perhaps (depending on data noise) also be assessed for their physiological decoders of increasing generality (Fig3e).

2. Testing the hypothesis of the general decoder:

2.1 In the data, the authors compare mainly the specific (stimulus) decoder and the monkey's choice decoder. The general stimulus decoder is only considered in Figure 3f, because data across multiple orientations are available only for the cued condition, and therefore the general and specific decoders cannot be compared for changes between cued and uncued. Fair enough, though this could be stated more explicitly around Line 160. However, the hypothesized relation between mean correlations and performance should also be true within a fixed attention condition (cued), comparing sessions with larger vs. smaller correlation. In other words, if the hypothesis is correct, you should find that performance of the "most general" decoder (as in Figure 3f) correlates negatively with average noise correlations, across sessions, more so than the "most specific" decoder. If there is enough data to identify this trend, it could strengthen the conclusions, because Figure 3c per se is not particularly overwhelming, and this reviewer is not sure that the correlation is significant for cued alone or uncued alone data, despite the fact that the range of noise correlation values within each condition is comparable to the range across conditions.

2.2 The analysis in fig3F provides a strong second line of argument in favor of the hypothesis. However, a lot hangs on the two points for "1 ori". This is because the authors restricted analysis to using the 4th orientation as a reference. It seems possible that this analysis could be repeated for other orientations apart from the 3rd on which the monkey data is trained so as to augment the number of points and bring out the relation more clearly.

2.3 In figure 3f, a more straightforward and precise comparison is to use the stimulus decoders to predict the choice, and test whether the more specific or the more general can predict choices more accurately.

3. The main goal of the manuscript is to determine the impact of noise correlations on various decoding schemes. The figures however only show how decoding co-varies with correlations, but a direct, more causal analysis of the effect of correlations on decoding seems to be missing. Such an analysis can be obtained by comparing decoding on simultaneously recorded activity with decoding on trial-shuffled activity, in which noise-correlations are removed. Related to this, the manuscript starts by stating that theoretical studies predict optimal decoding is independent of correlations. Yet, it is apparently never shown (using shuffles) that this prediction actually holds for the "specific" decoder. Conversely, it is not shown that the monkeys' or the general decoder are actually sensitive to removing correlations by shuffling.

4. Figure 3a: Why is the performance of the "monkey's decoder" so low on uncued trials? Is this fully explained by a change in the magnitude of correlations (ie would shuffling single trials in the cued trials lead to such a large decrease)? Or is it due to some other mechanism?

On a related note, how different are the four different decoders (specific/monkey, cued/uncued)? It would be interesting to see how much they overlap. More generally, the authors should discuss the alternative that attention modulates also the readout/decoding weights, rather than or in addition to modulating V4 activity. And also, that the decoder is just suboptimal, not suboptimal locally because optimized for generality. For instance, Figure 3a suggests that in the uncued condition there is lots of information in the neural activity, but the monkeys do not use it. In contrast, the general decoder in the model extracts a large fraction of the information (Figure 3b).

5. Quantifying the link between model and data:

5.1 the text providing motivation for the model could be improved. The motivation used in the manuscript is, essentially, that the model allows to extrapolate beyond the data (more stimuli, more repetitions, more neurons). That sounds weak, as the dangers of extrapolation beyond the range of the data are well known. A model that extrapolates beyond existing data is useful to design new experiments and test predictions, but this is not done here. Because the manuscript is about information and decoding, a better motivation is the fact that this model takes an actual image as input and produces tuning and covariance compatible with each other because they are constrained by an actual network that processes the input (as opposed to parametric models where tuning and covariance can be manipulated independently).

5.2 The ring structure, and the orientation of correlations (Figure 2b) seem to be key ingredients of the model, but are they based on data, or ad-hoc assumptions? L'179-181:"we first mapped the neuronal activity to a 2d space" – how was this done? What are the axes in Figure 2b? The correlation structure appears to be organized in 2d, how can one then understand the 1d changes in Figure 2f?

5.3 In the model, the specific decoder is quite strongly linked to correlated variability and the improvement of the general decoder is clear but incremental (0.66 vs 0.83) whereas in the data there really is no correlation at all (Figure 3c). This is a bit problematic because the authors begin by stating that specific decoders cannot explain the link between noise correlations and accuracy but their specific decoder clearly shows a link.

It may be that the comparison is a bit unfair on the author's hypothesis precisely because of the huge power provided by the model. In order to compare the magnitude of the effect with physiological data, the authors could down sample the results from their model to the range of correlated variability from the monkey data. This may be revealing because the correlation does not seem quite linear and plateaus out close to the monkey's range.

5.4 Quantitative mismatch between model and data: the model is intended to offer only qualitative predictions, and this is fine. But the reviewers did not understand the argument (eg. Line 191 and Line 320) that a quantitative mismatch is a good thing… after all, if the range of changes in noise correlations is small for the data, isn't that the relevant range?

6. General decoder: Some parts of the text (eg. Line 60, Line 413) refer to a decoder that accounts for discrimination along different stimulus dimensions (eg. different values of orientation, or different color of the visual input). But the results of the manuscripts are about a general decoder for multiple values along a single stimulus dimension. The disconnect should be discussed, and the relation between these two scenarios explained.

7. Some statements in the discussion such as l 354 "the relationship between behavior and mean correlated variability is explained by the hypothesis that observers use a general strategy" should be qualified: the authors clearly show that the general decoder amplifies the relationship but in their own data the relationship exists already with a specific decoder.

8. Low-Dimensionality, beginning of Introduction and end of Discussion: experimentally, cortical activity is low-dimensional, and the proposed model captures that. But this reviewer does not understand the argument offered for why this matters for the relation between average correlations and performance. It seems that the dimensionality of the population covariance is not relevant: The point instead is that a change in amplitude of fluctuations along the f'f' direction necessarily impact performance of a "specific" decoder, whereas changes in all other dimensions can be accounted for by the appropriate weights of the "specific" decoder. On the other hand, changes in fluctuation strength along multiple directions may impact the performance of the "general" decoder. Please revise the text to clarify.

[Editors' note: further revisions were suggested prior to acceptance, as described below.]

Thank you for resubmitting your work entitled "A general decoding strategy explains the relationship between behavior and correlated variability" for further consideration by *eLife*. Your revised article has been evaluated by Tirin Moore (Senior Editor) and a Reviewing Editor.

The manuscript has been improved but there are some remaining issues that need to be addressed, as outlined below.

The authors declined to investigate the impact of shuffling data on their results and the reason given (they are not making claims whether correlations vs no correlation is better) doesn't seem relevant to the point raised. The key issue is that the text suggests a causal, mechanistic link between correlations and the decoder, but this need not to be the case. For instance, in the model, the authors manipulate noise correlations via the modulation of simulated top-down feedback. This may impact other aspects of network activity rather than only correlations, and these other aspects may be responsible for the modified decoding. Similarly, changes in attention levels may indirectly lead to changes in both correlations and decoding, without the two being in a direct causal relation.

It seems like an easy and straight-forward sanity check to see if the accuracy of the two decoders correlates with attention level after shuffling both training and test sets. If shuffling has no effect on the results, the causal statements would need to be amended and/or discussed.

---

## [Author Response]

Essential revisions:A highly robust result when investigating how neural population activity is impacted by performance in a task is that the trial to trial correlations (noise correlations) between neurons is reduced as performance increases. However, the theoretical and experimental literature so far has failed to account for this robust link since reduced noise correlations do not systematically contribute to improved availability or transmission of information (often measured using decoding of stimulus identity). This paper sets out to address this discrepancy by proposing that the key to linking noise correlations to decoding and thus bridging the gap with performance is to rethink the decoders we use : instead of decoders optimized to the specific task imposed on the animal on any given trial (A vs B / B vs C / A vs C), they hypothesize that we should favor a decoder optimized for a general readout of stimulus properties (A vs B vs C).To test this hypothesis, the authors use a combination of quantitative data analysis and mechanistic network modeling. Data were recorded from neuronal populations in area V4 of two monkeys trained to perform an orientation change detection task, where the magnitude of orientation change could vary across trials, and the change could happen at cued (attended) or uncued (unattended) locations in the visual field. The model, which extends previous work by the authors, reproduces many basic features of the data, and both the model and data offer support (with one exception, details below) for the hypothesis.The reviewers agreed that this is a potentially important contribution, that addresses a widely observed, but puzzling, relation between perceptual performance and noise correlations. The clarity of the hypothesis, and the combination of data analysis and computational modelling are two essential strengths of the paper. Nonetheless, as detailed below, the reviewers believe the manuscript clarity could be further improved in several points, and some additional analysis of the data would provide more straightforward test of the hypothesis.Essential revisions:1. it would be important to verify that the model reproduces the correlation between noise and signal correlations since this is really a key argument leading to the author's hypothesis. One possibility would be to make a scatterplot of these two correlations for all neurons for both neural data and the model and for example compare slopes. The slope for the model could be shown for a range of levels of attentional modulation and for the neural data in attended vs unattended as they already do in fig2d of Fig2e. The authors could provide insight into the difference between the specific and general decoder by directly assessing the alignment between the decoders and the noise dimension. This could perhaps (depending on data noise) also be assessed for their physiological decoders of increasing generality (Fig3e).

We agree and we have incorporated this verification of the model into the manuscript. We modified the Results text as below to describe the new analysis and figures (lines 209 – 218 of the redlined manuscript; please see the redlined manuscript for all additions/deletions marked in red/strikethrough):

“Importantly, this model reproduces the correlation between noise and signal correlations (Figure 2—figure supplement 1) observed in electrophysiological data (Cohen and Maunsell, 2009; Cohen and Kohn, 2011). This correlation between the shared noise and the shared tuning is a key component of the general decoder hypothesis. We observed this strong relationship between noise and signal correlations in our recorded neurons (Figure 2—figure supplement 1A) as well as in our modeled data (Figure 2—figure supplement 1B). Using this model, we were able to measure the relationship between noise and signal correlations for varying strengths of attentional modulation. Consistent with the predictions of the general decoder hypothesis, attention weakened the relationship between noise and signal correlations (Figure 2—figure supplement 1C).”

As predicted by our published simulations (Ruff, Ni, and Cohen, 2019), neither the model nor the electrophysiological data provided insights into the difference between the specific and general decoder through a direct assessment of the alignment between the decoders and the noise dimension. The intuition is that recording from small subsets of a population provides only weak constraints on the actual weightings of each neuron (which determine the decoding dimensions). We therefore test predictions of these dimensions such as these relationships between signal and noise correlations and the decoding performance using different decoding strategies.

2. Testing the hypothesis of the general decoder:2.1 In the data, the authors compare mainly the specific (stimulus) decoder and the monkey's choice decoder. The general stimulus decoder is only considered in Figure 3f, because data across multiple orientations are available only for the cued condition, and therefore the general and specific decoders cannot be compared for changes between cued and uncued. Fair enough, though this could be stated more explicitly around Line 160. However, the hypothesized relation between mean correlations and performance should also be true within a fixed attention condition (cued), comparing sessions with larger vs. smaller correlation. In other words, if the hypothesis is correct, you should find that performance of the "most general" decoder (as in Figure 3f) correlates negatively with average noise correlations, across sessions, more so than the "most specific" decoder. If there is enough data to identify this trend, it could strengthen the conclusions, because Figure 3c per se is not particularly overwhelming, and this reviewer is not sure that the correlation is significant for cued alone or uncued alone data, despite the fact that the range of noise correlation values within each condition is comparable to the range across conditions.

We appreciate this idea, but we did not have enough data to determine if the most general decoder that we could calculate with our current electrophysiological dataset was more negatively correlated with average noise correlations than our most specific decoder. We modified the manuscript to include additional analyses of Figure 3C within each individual attention condition, now illustrated in Figure 3—figure supplement 1. We also modified the manuscript to clarify what we could and could not address using our electrophysiological data alone, and to be more explicit about the critical role of our modeled data in addressing our hypothesis (this point is addressed further below in response to point 5.1). The manuscript modifications that address these points are noted below.

New text in the Results section (lines 361 – 364):

“Indeed, we found that just as the performance of the physiological monkey’s decoder was more strongly related to mean correlated variability than the performance of the physiological specific docoder (Figure 3C; see Figure 3—figure supplement 1 for analyses per attention condition)…”

New text in the Results section to clarify the limits of the electrophysiological data and the importance of our modeled data (lines 173 – 187):

“For our electrophysiological dataset, the behavioral task was designed to allow us to compare the specific and monkey’s decoders for an attention task with a range of orientation change amounts. To collect the necessary number of repetitions of behavioral trials per stimulus condition (with the limited total number of behavioral trials collected per day), we limited the number of different orientation change amounts to five (Figure 1B) and focused our uncued trials on the median orientation change. Our modeled dataset is critical to addressing our general decoder hypothesis as it allows us to step beyond the restraints of physiological data to model multiple attentional modulation levels for the full range of stimulus orientations. While the main purpose of the electrophysiological data was to analyze the monkey’s decoder, which could only be determined using the neuronal responses recorded from a behaving animal, the purpose of our modeled data is to compare the monkey’s decoder to an ideal general decoder, which can only be determined here using a model.”

Modified text in the Result section to state more explicitly that the more-general decoders could not be calculated for the uncued condition (lines 382 – 390):

“Finally, while the circuit model allowed us to analyze an ideal general decoder for the full range of stimulus orientations in multiple attention conditions (Figure 2B, C; a full ring of orientations in both unattended and attended conditions), as a sanity check, we used the physiological data from the cued attention condition only (a limited set of five orientation change amounts as illustrated in Figure 1B, in the cued condition only as the uncued condition focused on the median orientation change amount only) to test whether more-general decoders were more related to the monkey’s decoder than more-specific decoders. Would a decoder based on more stimulus orientations be more related to the monkey’s decoder than a decoder based on fewer stimulus orientations?”

2.2 The analysis in fig3F provides a strong second line of argument in favor of the hypothesis. However, a lot hangs on the two points for "1 ori". This is because the authors restricted analysis to using the 4th orientation as a reference. It seems possible that this analysis could be repeated for other orientations apart from the 3rd on which the monkey data is trained so as to augment the number of points and bring out the relation more clearly.

We thank the reviewers for this idea and we have replaced our original Figure 3F with an updated plot that follows this suggestion exactly, performing the analysis for each of the other orientations apart from the third orientation:

We have updated the Methods to describe the new analysis, as below (lines 701 – 735):

“The most specific decoder tested in Figure 3F was trained to best differentiate the V4 population responses to one changed orientation from the V4 responses to the starting orientation presented immediately before it. The performance of this specific decoder was quantified as the leave-one-out cross-validated proportion of trials on which the decoder correctly identified whether the left-out orientation was that one changed orientation or the starting orientation. The one changed orientation that was tested was either the first, second, fourth, or fifth largest changed orientation (all but the median changed orientation, which was excluded from this analysis due to the inherent relationship between any decoder that included this orientation and the monkey’s decoder which was based on this orientation, as noted above). We calculated the performance of each decoder on each day of recording. We then calculated the across-days correlation between the performance of this specific decoder and the performance of the monkey’s decoder (described above). The correlation coefficient was plotted in the ‘1 ori’ column of Figure 3F for each monkey (*n* = 8 decoders; 1 decoder for each of the 4 included changed orientations, for each of the 2 monkeys).

We performed this same procedure for increasingly more-general decoders. The decoders illustrated in the ‘2 oris’ column were trained to best differentiate the V4 responses to two changed orientations from the V4 responses to the starting orientation. The two changed orientations used for each ‘2 oris’ decoder were chosen from the four possibilities: the first, second, fourth, and fifth (max) largest changed orientations (*n* = 48 decoders; 1 decoder for each of the 6 combinations of 2 changed orientations, tested on each of the 4 included changed orientations, for each of the 2 monkeys). We calculated the performance of each ‘2 oris’ decoder on the same task as above: identifying whether the orientation was the changed orientation (the first, second, fourth, or fifth largest changed orientation) or the starting orientation. Again, we calculated the across-days correlation between the performance of each ‘2 oris’ decoder and the performance of the monkey’s decoder.

Each ‘3 oris’ decoder was trained to best differentiate the V4 responses to three changed orientations from the V4 responses to the starting orientation (*n* = 32 decoders; 1 decoder for each of the 4 combinations of 3 changed orientations, tested on each of the 4 included changed orientations, for each of the 2 monkeys). The ‘4 oris’ decoder for each monkey was trained to best differentiate the V4 responses to four changed orientations from the V4 responses to the starting orientation (*n* = 8 decoders; 1 decoder for the 1 combination of 4 changed orientations, tested on each of the 4 included changed orientations, for each of the 2 monkeys).”

2.3 In figure 3f, a more straightforward and precise comparison is to use the stimulus decoders to predict the choice, and test whether the more specific or the more general can predict choices more accurately.

We thank the reviewers for this suggestion and have added a new figure (Figure 3G) that illustrates the results of this analysis comparing whether the specific or more-general decoders predict the monkey’s trial-by-trial choices more accurately:

The Figure 3G legend is as below (lines 343 – 347):

“Figure 3… (G) The more general the decoder (*x*-axis), the better its performance predicting the monkey’s choices on the median changed orientation trials (*y*-axis; the proportion of leave-one-out trials in which the decoder correctly predicted the monkey’s decision as to whether the orientation was the starting orientation or the median changed orientation). Conventions as in (F) (see Methods for *n* values).”

We added the following to the Results section (lines 402 – 404):

“Further, the more general the decoder, the better it predicted the monkey’s trial-by-trial choices on the median changed orientation trials (Figure 3G).”

We have updated the Methods section of the manuscript to describe this modified analysis, as below (lines 736 – 749):

“For Figure 3G, we performed analyses similar to those performed for Figure 3F, in that we tested each stimulus decoder: ‘1 ori’ decoders (*n* = 8 decoders; 1 specific decoder for either the first, second, fourth, or fifth largest changed orientation, for each of the 2 monkeys), ‘2 oris’ decoders (*n* = 12 decoders; 1 decoder for each of the 6 combinations of 2 changed orientations, for each of the 2 monkeys), ‘3 oris’ decoders (*n* = 8 decoders; 1 decoder for each of the 4 combinations of 3 changed orientations, for each of the 2 monkeys), and ‘4 oris’ decoders (*n* = 2 decoders; 1 decoder for the 1 combination of 4 changed orientations, for each of the 2 monkeys). However, unlike in Figure 3F, where the performance of the stimulus decoders was compared to the performance of the monkey’s decoder on the median orientation-change trials, here we calculated the performance of the stimulus decoder when tasked with predicting the trial-by-trial choices that the monkey made on the median orientation-change trials. We plotted the proportion of leave-one-out trials in which each decoder correctly predicted the monkey’s choice as to whether the orientation was the starting orientation or the median changed orientation.”

3. The main goal of the manuscript is to determine the impact of noise correlations on various decoding schemes. The figures however only show how decoding co-varies with correlations, but a direct, more causal analysis of the effect of correlations on decoding seems to be missing. Such an analysis can be obtained by comparing decoding on simultaneously recorded activity with decoding on trial-shuffled activity, in which noise-correlations are removed. Related to this, the manuscript starts by stating that theoretical studies predict optimal decoding is independent of correlations. Yet, it is apparently never shown (using shuffles) that this prediction actually holds for the "specific" decoder. Conversely, it is not shown that the monkeys' or the general decoder are actually sensitive to removing correlations by shuffling.

We understand the spirit of this suggestion. But, we do feel that the goal of this study is to understand implications of the relationship between noise correlations and decoder performance. We purposely do not make claims about the utility of noise correlations relative to hypothetical situations in which there are no correlations, because we suspect those are physiologically impossible (and are also impossible in the current incarnation of our model).

We have added the following Discussion section to address this point (lines 525 – 536):

“The purpose of this study was to investigate the relationship between mean correlated variability and a general decoder. We made an initial test of the overarching hypothesis that observers use a general decoding strategy in feature-rich environments by testing whether a decoder optimized for a broader range of stimulus values better matched the decoder actually used by the monkeys than a specific decoder optimized for a narrower range of stimulus values. We purposefully did not make claims about the utility of correlated variability relative to hypothetical situations in which correlated variability does not exist in the responses of a group of neurons, as we suspect that this is not a physiologically realistic condition. Studies that causally manipulate the level of correlated variability in neuronal populations to measure the true physiological and behavioral effects of increasing or decreasing correlated variability levels, through pharmacological or genetic means, may provide important insights into the impact of correlated variability on various decoding strategies.”

4. Figure 3a: Why is the performance of the "monkey's decoder" so low on uncued trials? Is this fully explained by a change in the magnitude of correlations (ie would shuffling single trials in the cued trials lead to such a large decrease)? Or is it due to some other mechanism?On a related note, how different are the four different decoders (specific/monkey, cued/uncued)? It would be interesting to see how much they overlap. More generally, the authors should discuss the alternative that attention modulates also the readout/decoding weights, rather than or in addition to modulating V4 activity. And also, that the decoder is just suboptimal, not suboptimal locally because optimized for generality. For instance, Figure 3a suggests that in the uncued condition there is lots of information in the neural activity, but the monkeys do not use it. In contrast, the general decoder in the model extracts a large fraction of the information (Figure 3b).

We have added the section below to the Discussion to better address these points, and have noted other possible explanations for the suboptimality of the monkey’s decoder (lines 428 – 447):

“A fixed readout mechanism

A prior study from our lab found that attention, rather than changing the neuronal weights of the observer’s decoder, reshaped neuronal population activity to better align with a fixed readout mechanism (Ruff and Cohen, 2019). To test whether the neuronal weights of the monkey’s decoder changed across attention conditions (attended versus unattended), Ruff and Cohen switched the neuronal weights across conditions, testing the stimulus information in one attention condition with the neuronal weights from the other. They found that even with the switched weights, the performance of the monkey’s decoder was still higher in the attended condition. The results of this study support the conclusion that attention reshapes neuronal activity so that a fixed readout mechanism can better read out stimulus information. In other words, differences in the performance of the monkey’s decoder across attention conditions may be due to differences in how well the neuronal activity aligns with a fixed decoder.

Our study extends the findings of Ruff and Cohen to test whether that fixed readout mechanism is determined by a general decoding strategy. Our findings support the hypothesis that observers use a general decoding strategy in the face of changing stimulus and task conditions. Our findings do not exclude other potential explanations for the suboptimality of the monkey’s decoder, nor do they exclude the possibility that attention modulates decoder neuronal weights. However, our findings together with those of Ruff and Cohen shed light on why neuronal decoders are suboptimal in a manner that aligns the fixed decoder axis with the correlated variability axis (Ni et al., 2018; Ruff et al., 2018).”

The section above references the following paper: Ruff DA and Cohen MR (2019) *Nat Neurosci* 22:1669-1676. Ruff and Cohen found similar levels of cued and uncued performance for area MT stimulus decoders and monkey decoders as the levels we found in area V4 (their Figure 3A) and concluded that the monkey decoder’s performance was so low in the uncued versus cued condition because attention reshapes neuronal activity so that more of the existing stimulus information can be used to guide behavior (their Figure 3B; also see their Figure 4B for similar results in V4).

5. Quantifying the link between model and data:5.1 the text providing motivation for the model could be improved. The motivation used in the manuscript is, essentially, that the model allows to extrapolate beyond the data (more stimuli, more repetitions, more neurons). That sounds weak, as the dangers of extrapolation beyond the range of the data are well known. A model that extrapolates beyond existing data is useful to design new experiments and test predictions, but this is not done here. Because the manuscript is about information and decoding, a better motivation is the fact that this model takes an actual image as input and produces tuning and covariance compatible with each other because they are constrained by an actual network that processes the input (as opposed to parametric models where tuning and covariance can be manipulated independently).

We appreciate this point and have updated our manuscript as below.

We added the following section in the Results (lines 161 – 172):

“Here, we describe a circuit model that we designed to allow us to compare the specific and monkey’s decoders from our electrophysiological dataset to modeled ideal specific and general decoders. The primary benefit of our model is that it can take actual images as inputs and produce neuronal tuning and covariance that are compatible with each other because of constraints from the simulated network that processed the inputs (Huang et al., 2019). Parametric models in which tuning and covariance can be manipulated independently would not provide such constraints. In our model, the mean correlated variability of the population activity is restricted to very few dimensions, matching experimentally recorded data from visual cortex demonstrating that mean correlated variability occupies a low-dimensional subset of the full neuronal population space (Ecker et al., 2014; Goris et al., 2014; Huang et al., 2019; Kanashiro et al., 2017; Lin et al., 2015; Rabinowitz et al., 2015; Semedo et al., 2019; Williamson et al., 2016).”

We also removed the Results sections that suggested that the model allowed us to extrapolate beyond the data.

We modified the following section of the Discussion (lines 417 – 423):

“Our study also demonstrates the utility of combining electrophysiological and circuit modeling approaches to studying neural coding. Our model mimicked the correlated variability and effects of attention in our physiological data. Critically, our model produced neuronal tuning and covariance based on the constraints of an actual network capable of processing images as inputs. Using a circuit model allowed us to test a full range of stimulus orientations in multiple attention conditions, allowing us to test the effects of attention on a true general decoder for orientation.”

We also removed the Discussion section that suggested that a benefit of the model was extrapolating beyond the data.

5.2 The ring structure, and the orientation of correlations (Figure 2b) seem to be key ingredients of the model, but are they based on data, or ad-hoc assumptions? L'179-181:"we first mapped the neuronal activity to a 2d space" – how was this done? What are the axes in Figure 2b? The correlation structure appears to be organized in 2d, how can one then understand the 1d changes in Figure 2f?

We have modified the manuscript to clarify the above points, as below.

In the Results section (lines 198 – 208):

“As the basis for our modeled general decoder, we first mapped the *n*-dimensional neuronal activity of our model in response to the full range of orientations to a 2-dimensional space. Because the neurons were tuned for orientation, we could map the *n*-dimensional population responses to a ring (Figure 2B, C). The orientation of correlations (the shape of each color cloud in Figure 2B) was not an assumed parameter, and illustrates the outcome of the correlation structure and dimensionality modeled by our data. In Figure 2B, we can see that the fluctuations along the radial directions are much larger than those along other directions for a given orientation. This is consistent with the low-dimensional structure of the modeled neuronal activity. In our model, the fluctuations of the neurons, mapped to the radial direction on the ring, were more elongated in the unattended state (Figure 2B) than in the attended state (Figure 2C).”

The mapping was done by minimizing the distance between a linear readout, w^T^ r, and a position on a ring with radius 1, (cos(theta), sin(theta)), for a given orientation theta. In the Methods section (lines 804 – 808):

We have modifed the Figure 2 legend and added that the axes in Figure 2B and C are arbitrary units (lines 242 – 248):

“Figure 2…(B, C) We mapped the *n*-dimensional neuronal activity of our model to a 2-dimensional space (a ring). Each dot represents the neuronal activity of the simulated population on a single trial and each color represents the trials for a given orientation. These fluctuations are more elongated in the (B) unattended state than in the (C) attended state. We then calculated the effects of these attentional changes on the performance of specific and general decoders (see Methods). The axes are arbitrary units.”

5.3 In the model, the specific decoder is quite strongly linked to correlated variability and the improvement of the general decoder is clear but incremental (0.66 vs 0.83) whereas in the data there really is no correlation at all (Figure 3c). This is a bit problematic because the authors begin by stating that specific decoders cannot explain the link between noise correlations and accuracy but their specific decoder clearly shows a link.It may be that the comparison is a bit unfair on the author's hypothesis precisely because of the huge power provided by the model. In order to compare the magnitude of the effect with physiological data, the authors could down sample the results from their model to the range of correlated variability from the monkey data. This may be revealing because the correlation does not seem quite linear and plateaus out close to the monkey's range.

We appreciate this point and we have modified our manuscript to clarify that our focus is on whether the general decoder is more strongly linked to correlated variability than the specific decoder. We have modified the manuscript as below.

In the Results (lines 361 – 381):

“Indeed, we found that just as the performance of the physiological monkey’s decoder was more strongly related to mean correlated variability than the performance of the physiological specific docoder (Figure 3C; see Figure 3—figure supplement 1 for analyses per attention condition), the performance of the modeled general decoder was more strongly related to mean correlated variability than the performance of the modeled specific decoder (Figure 3D). We modeled much stronger relationships to correlated variability (Figure 3D) than observed with our physiological data (Figure 3C). We observed that the correlation with specific decoder performance was significant with the modeled data but not with the physiological data. This is not surprising as we saw attentional effects, albeit small ones, on specific decoder performance with both the physiological and the modeled data (Figure 3A, B). Even small attentional effects would result in a correlation between decoder performance and mean correlated variability with a large enough range of mean correlated variability values. It is possible that with enough electrophysiological data, the performance of the specific decoder would be significantly related to correlated variability, as well. As described above, our focus is not on whether the performance of any one decoder is significantly correlated with mean correlated variability, but on which decoder provides a better explanation of the frequently observed relationship between performance and mean correlated variability. The performance of the general decoder was more strongly related to mean correlated variability than the performance of the specific decoder.”

In the Discussion (lines 407 – 409):

“Our results suggest that the relationship between behavior and mean correlated variability is more consistent with observers using a more general strategy that employs the same neuronal weights for decoding any stimulus change.”

Additionally, we have analyzed just two attention conditions from Figure 3D and have plotted the results in Author response image 1. We are happy to add Author response image 1 to the manuscript, but have not included it for now as we believe that our adjusted manuscript texts above better highlight that our goal was not to make a match between the electrophysiological and modeled datasets, but to illustrate that the general decoder is more strongly related to correlated variability than the specific decoder. In Author response image 1 are the results of our model analysis based on two attention conditions only:

**Author response image 1. sa2fig1:** 

5.4 Quantitative mismatch between model and data: the model is intended to offer only qualitative predictions, and this is fine. But the reviewers did not understand the argument (eg. Line 191 and Line 320) that a quantitative mismatch is a good thing… after all, if the range of changes in noise correlations is small for the data, isn't that the relevant range?

We agree with this point and we have modified the manuscript to focus on the primary benefits of the model. We have made the following changes.

We removed the section of text that argued that the quantitative mismatch was a good thing.

We removed the text that described the benefits of a quantitative mismatch.

6. General decoder: Some parts of the text (eg. Line 60, Line 413) refer to a decoder that accounts for discrimination along different stimulus dimensions (eg. different values of orientation, or different color of the visual input). But the results of the manuscripts are about a general decoder for multiple values along a single stimulus dimension. The disconnect should be discussed, and the relation between these two scenarios explained.

We thank the reviewers for this helpful suggestion. We have modified the manuscript to better clarify the relationship between our current study, which is an initial test of the general decoder hypothesis based on a single stimulus dimension, and the overarching idea of a general decoder for multiple stimulus dimensions.

In the Introduction (lines 71 – 82):

“Here, we report the results of an initial test of this overarching hypothesis, based on a single stimulus dimension. We used a simple, well-studied behavioral task to test whether a more-general decoder (optimized for a broader range of stimulus values along a single dimension) better explained the relationship between behavior and mean correlated variability than a more-specific decoder (optimized for a narrower range of stimulus values along a single dimension). Specifically, we used a well-studied orientation change-detection task (Cohen and Maunsell, 2009) to test whether a general decoder for the full range of stimulus orientations better explained the relationship between behavior and mean correlated variability than a specific decoder for the orientation change presented in the behavioral trial at hand.

This test based on a single stimulus dimension is an important initial test of the general decoder hypothesis because many of the studies that found that performance increased when mean correlated variability decreased used a change-detection task…”

In the Discussion (lines 450 – 459):

“We performed this initial test of the overarching general decoder hypothesis in the context of a change-detection task along a single stimulus dimension because this type of task was used in many of the studies that reported a relationship between perceptual performance and mean correlated variability (Cohen and Maunsell, 2009; 2011; Herrero et al., 2013; Luo and Maunsell, 2015; Mayo and Maunsell, 2016; Nandy et al., 2017; Ni et al., 2018; Ruff and Cohen, 2016; 2019; Verhoef and Maunsell, 2017; Yan et al., 2014; Zénon and Krauzlis, 2012). This simple and well-studied task provided an ideal initial test of our general decoder hypothesis.

This initial test of the general decoder hypothesis suggests that a more general decoding strategy may explain observations in studies that use a variety of behavioral and stimulus conditions.”

In the Discussion (lines 511 – 524):

“This initial study of the general decoder hypothesis tested this idea in the context of a visual environment in which stimulus values only changed along a single dimension. However, our overarching hypothesis is that observers use a general decoding strategy in the complex and feature-rich visual scenes encountered in natural environments. In everyday environments, visual stimuli can change rapidly and unpredictably along many stimulus dimensions. The hypothesis that such a truly general decoder explains the relationship between perceptual performance and mean correlated variability is suggested by our finding that the modeled general decoder for orientation was more strongly related to mean correlated variability than the modeled specific decoder (Figure 3D). Future tests of a general decoder for multiple stimulus features would be needed to determine if this decoding strategy is used in the face of multiple changing stimulus features. Further, such tests would need to consider alternative hypotheses for how sensory information is decoded when observing multiple aspects of a stimulus (Berkes et al., 2009; Deneve, 2012; Lorteije et al., 2015). Studies that use complex or naturalistic visual stimuli may be ideal for further investigations of this hypothesis.”

7. Some statements in the discussion such as l 354 "the relationship between behavior and mean correlated variability is explained by the hypothesis that observers use a general strategy" should be qualified: the authors clearly show that the general decoder amplifies the relationship but in their own data the relationship exists already with a specific decoder.

We have updated the Discussion section of the manuscript in several places to better qualify our conclusions, as described below.

In lines 407 – 409 (previously line 354 – 356):

“Our results suggest that the relationship between behavior and mean correlated variability is more consistent with observers using a more general strategy that employs the same neuronal weights for decoding any stimulus change.

In lines 414 – 416:

“Together, these results support the hypothesis that observers use a more general decoding strategy in scenarios that require flexibility to changing stimulus conditions.”

In lines 457 – 459:

“This initial test of the general decoder hypothesis suggests that a more general decoding strategy may explain observations in studies that use a variety of behavioral and stimulus conditions.”

8. Low-Dimensionality, beginning of Introduction and end of Discussion: experimentally, cortical activity is low-dimensional, and the proposed model captures that. But this reviewer does not understand the argument offered for why this matters for the relation between average correlations and performance. It seems that the dimensionality of the population covariance is not relevant: The point instead is that a change in amplitude of fluctuations along the f'f' direction necessarily impact performance of a "specific" decoder, whereas changes in all other dimensions can be accounted for by the appropriate weights of the "specific" decoder. On the other hand, changes in fluctuation strength along multiple directions may impact the performance of the "general" decoder. Please revise the text to clarify.

We appreciate this point and have updated the manuscript Introduction and Discussion to better explain our motivation for using a low-dimensional model. We clarify that our low-dimensional model is beneficial because experimentally recorded cortical activity is low-dimensional. Our changes are as below.

We modified the following text in the Introduction (lines 44 – 55):

“These observations comprise a paradox because changes in this simple measure should have a minimal effect on information coding. Recent theoretical work shows that neuronal population decoders that extract the maximum amount of sensory information for the specific task at hand can easily ignore mean correlated noise (Kafashan et al., 2021; Kanitscheider et al., 2015b; Moreno-Bote et al., 2014; Pitkow et al., 2015; Rumyantsev et al., 2020; for review, see Kohn et al., 2016). Decoders for the specific task at hand can ignore mean correlated variability because it does not corrupt the dimensions of neuronal population space that are most informative about the stimulus (Moreno-Bote et al., 2014).”

We added the following text describing that the model captures the low-dimensional nature of cortical activity, thus placing the emphasis on the fact that the model captures the low dimensionality, instead of on the relevance of this low dimensionality to the relationship between average correlations and performance (lines 163 – 172):

“The primary benefit of our model is that it can take actual images as inputs and produce neuronal tuning and covariance that are compatible with each other because of constraints from the simulated network that processed the inputs (Huang et al., 2019). Parametric models in which tuning and covariance can be manipulated independently would not provide such constraints. In our model, the mean correlated variability of the population activity is restricted to very few dimensions, matching experimentally recorded data from visual cortex demonstrating that mean correlated variability occupies a low-dimensional subset of the full neuronal population space (Ecker et al., 2014; Goris et al., 2014; Huang et al., 2019; Kanashiro et al., 2017; Lin et al., 2015; Rabinowitz et al., 2015; Semedo et al., 2019; Williamson et al., 2016).”

We modified the end of the Discussion as follows (lines 481 – 492):

“Our results address a paradox in the literature. Electrophysiological and theoretical evidence supports that there is a relationship between mean correlated variability and perceptual performance (Abbott and Dayan, 1999; Clery et al., 2017; Haefner et al., 2013; Jin et al., 2019; Ni et al., 2018; Ruff and Cohen, 2019; reviewed by Ruff et al., 2018). Yet, a specific decoding strategy in which different sets of neuronal weights are used to decode different stimulus changes cannot easily explain this relationship (Kafashan et al., 2021; Kanitscheider et al., 2015b; Moreno-Bote et al., 2014; Pitkow et al., 2015; Rumyantsev et al., 2020; reviewed by Kohn et al., 2016). This is because specific decoders of neuronal population activity can easily ignore changes in mean correlated noise (Moreno-Bote et al., 2014).”

[Editors' note: further revisions were suggested prior to acceptance, as described below.]

The manuscript has been improved but there are some remaining issues that need to be addressed, as outlined below.The authors declined to investigate the impact of shuffling data on their results and the reason given (they are not making claims whether correlations vs no correlation is better) doesn't seem relevant to the point raised. The key issue is that the text suggests a causal, mechanistic link between correlations and the decoder, but this need not to be the case. For instance, in the model, the authors manipulate noise correlations via the modulation of simulated top-down feedback. This may impact other aspects of network activity rather than only correlations, and these other aspects may be responsible for the modified decoding. Similarly, changes in attention levels may indirectly lead to changes in both correlations and decoding, without the two being in a direct causal relation.It seems like an easy and straight-forward sanity check to see if the accuracy of the two decoders correlates with attention level after shuffling both training and test sets. If shuffling has no effect on the results, the causal statements would need to be amended and/or discussed.

We have performed the shuffling analysis described above and have added a new figure, Figure 3—figure supplement 2.

We have added text to the Results section to describe the new figure, as below:

“… the performance of the modeled general decoder was more strongly related to mean correlated variability than the performance of the modeled specific decoder (Figure 3D; see Figure 3—figure supplement 2 for trial-shuffled analyses).”

Finally, we have added text to the Discussion section to describe the new shuffling analyses, as below:

“In our model, which was designed to mimic real data, attention changed many aspects of neural responses besides just correlated variability. It is therefore possible that any relationship between decoding performance and correlated variability is mostly caused by those concomitant changes. Therefore, we used the many trials in our modeled data to test the effects of randomly shuffling the trial order per modeled neuron. These shuffled data resulted in the modeled general and specific decoders becoming essentially indistinguishable in their relationships with the removed correlated variability (Figure 3—figure supplement 2), with those removed correlations essentially representing attention condition. The effects of attention on many aspects of neuronal population activity have been well documented, including effects on neuronal firing rates and on both individual and shared trial-to-trial response variability (Cohen and Maunsell, 2009; 2011; Herrero et al., 2013; Luo and Maunsell, 2015; Mayo and Maunsell, 2016; Mitchell et al., 2009; Nandy et al., 2017; Ni et al., 2018; Ruff and Cohen, 2014a; 2014b; 2016; 2019; Zénon and Krauzlis, 2012). The simulated neurons in our model captured many of these attention effects (Figure 2D-F; Huang et al., 2019). Much theoretical work has supported that in the presence of correlations between neuronal responses, specifically differential or information-limiting correlations, limits on decoding performance will be dominated by these correlations (Moreno-Bote et al., 2014; for review, see Kohn et al., 2016). Removing these correlations by shuffling the trials results in many changes; in particular, our trial-shuffled data demonstrate the well-documented linear growth in Fisher information that is expected with increasing numbers of neurons (Figure 3—figure supplement 2A-C; Averbeck et al., 2006; Kohn et al., 2016; Shadlen et al., 1996). Our trial-shuffled analysis illustrates that removing correlations results in decoder performance being dominated by other effects of attention on neuronal activity, such as the firing rates (gains) of the neurons. Our model reproduces the gain effects of attention on neuronal firing rates observed in electrophysiological data (Figure 2D) which, in the absence of correlations, increases the sensitivity of the population (Averbeck et al., 2006; Kohn et al., 2016; Shadlen et al., 1996). In summary, general and specific decoder performances had indistinguishable relationships with the amount of correlated variability removed by the trial shuffling (Figure 3—figure supplement 2D, E), suggesting that decoder performance became dominated by attention-related firing rate gains intrinsic to our model.